# Cortex-wide neural interfacing via transparent polymer skulls

Leila Ghanbari [1], Russell E. Carter[2], Mathew L. Rynes [3], Judith Dominguez[1], Gang Chen[2], Anant Naik[3], Jia Hu[3], Md Abdul Kader Sagar[4], Lenora Haltom[1], Nahom Mossazghi[2], Madelyn M. Gray[2], Sarah L. West[2], Kevin W. Eliceiri[4], Timothy J. Ebner [2] & Suhasa B. Kodandaramaiah[1,3]

Neural computations occurring simultaneously in multiple cerebral cortical regions are critical for mediating behaviors. Progress has been made in understanding how neural activity in specific cortical regions contributes to behavior. However, there is a lack of tools that allow simultaneous monitoring and perturbing neural activity from multiple cortical regions. We engineered 'See-Shells'—digitally designed, morphologically realistic, transparent polymer skulls that allow long-term (>300 days) optical access to 45 mm$^2$ of the dorsal cerebral cortex in the mouse. We demonstrate the ability to perform mesoscopic imaging, as well as cellular and subcellular resolution two-photon imaging of neural structures up to 600 μm deep. See-Shells allow calcium imaging from multiple, non-contiguous regions across the cortex. Perforated See-Shells enable introducing penetrating neural probes to perturb or record neural activity simultaneously with whole cortex imaging. See-Shells are constructed using common desktop fabrication tools, providing a powerful tool for investigating brain structure and function.

[1] Department of Mechanical Engineering, University of Minnesota, Twin Cities, MN, USA. [2] Department of Neuroscience, University of Minnesota, Twin Cities, MN, USA. [3] Department of Biomedical Engineering, University of Minnesota, Twin Cities, MN, USA. [4] Department of Biomedical Engineering, University of Wisconsin, Madison, WI, USA. Correspondence and requests for materials should be addressed to S.B.K. (email: suhasabk@umn.edu)

The mammalian cerebral cortex mediates learned and adaptive forms of sensory–motor behaviors and the evolutionary expansion of the cortex underlies many advanced cognitive capabilities in humans and non-human primates. Neuroscientists have taken advantage of the modular organization and segregation of the cortex into anatomically and functionally distinct regions and have made enormous progress understanding how computations performed in specific cortical regions engage in behavior. However, operation of the brain cannot be understood only by analysis of its components in isolation. Yet, the mechanisms by which neural activity is coordinated across the cerebral cortex to produce a unitary behavioral output are not well understood. Even simple sensory or motor tasks involve processing of information in multiple cortical areas. For example, deflection of a single whisker results in activation distributed across the sensorimotor cortices[1], locomotion modulates the neural responses in the primary visual cortex with cell-type specificity[2], and arousal exerts markedly different effects across the cerebral cortex, both spatially and temporally[3]. Further, such long-range information flow is dependent on the internal brain state as well as information learned from past experiences.

Understanding these large-scale computations requires the ability to monitor and perturb neural activity across large regions of the cortex. Until recently, the study of mesoscopic and macroscopic brain networks has been limited to functional magnetic resonance imaging (fMRI) and magnetoencephalography (MEG). However, fMRI and MEG are limited by their spatial and temporal resolution. Two-photon (2P) imaging has rapidly emerged as a tool of choice for in vivo imaging in rodent models due to improved deep imaging over one-photon (1P) imaging methods[4]. The development of optical sensing tools in recent years allows 2P-based cellular resolution monitoring of neural activities in local circuits. Genetically encoded $Ca^{2+}$ indicators (such as GCaMP6) have enabled in vivo high-resolution monitoring of activities of hundreds to thousands of neurons[5]. The development of red-shifted variants of $Ca^{2+}$ indicators and optogenetic tools open deep cortical regions for optical sensing and perturbation[6,7]. The advent of streamlined strategies to rapidly generate transgenic mice expressing optical reporters has been matched by the recent development of wide-field 2P imaging approaches[8–10].

Deploying these new optical tools and instrumentation for chronic imaging of large areas of the cerebral cortex requires replacing the overlying opaque skull with a transparent substrate. A widely used method to achieve chronic optical access to the brain surface involves implanting optical windows or cranial windows in which sections of the skull are excised and replaced with glass coverslips[11]. To image even larger regions, strategies for refractive index matching[12] and thinned skull preparations[13] have been used. These techniques, however, do not reliably allow cellular resolution imaging as image quality is dependent on the surgical preparation and is susceptible to bone regrowth over time. Recently, curved glass windows and associated surgical implantation methodology were introduced that allow chronic optical access to the whole dorsal cortex for cellular resolution imaging[14]. While each of these approaches has advanced the field, each has limitations. Ideally, wide-field optical imaging would be combined with modalities that allow simultaneous perturbation of neural activities to reveal the effect of various brain regions on global cortical activity. Further, combining wide-field optical imaging with simultaneous electrophysiological recordings from different brain regions will provide a better understanding of how global activity patterns modulate activity in local circuits[15]. Therefore, large optical windows with excellent optical properties, long-term functionality, design flexibility, easy fabrication and surgical implantation, and accommodation of other modalities are needed.

Here we introduce See-Shells, digitally designed and morphologically realistic transparent polymer skulls that can be chronically implanted for long durations (>300 days) and allow optical access to 45 mm² of the dorsal cerebral cortex. See-Shells can be customized to fit a variety of skull morphologies and allow for sub-cellular resolution structural imaging. Further, $Ca^{2+}$ imaging can be performed at both mesoscale and cellular resolution from populations of neurons spread across millimeters of the cortex during awake, head-fixed behavior. See-Shells are easily adapted to include perforations for penetrating stimulation or recording probes. We also demonstrate the ability to perform wide-field $Ca^{2+}$ imaging simultaneously with intracortical microstimulation and extracellular recordings. See-Shells can be inexpensively fabricated using desktop prototyping tools and can be implanted using methodologies adapted from standard cranial window implantation procedures.

## Results

**Device design and fabrication.** The overall design of the See-Shells is shown in Fig. 1a. A motorized stereotaxic instrument was used to profile the skull surface covering the dorsal cortex of an 8-week-old C57BL/6 mouse at 85 points (see Methods). These 85 coordinates provided a point cloud representation of the skull surface used to interpolate a three-dimensional (3D) surface that accurately mimicked the skull morphology (Supplementary Figs. 1 and 2). Previous cranial morphometry studies of commonly used inbred laboratory mouse strains have shown that intra-species variations in cranial bone shape and size are minimal[16]. Further, postnatal size and shape of the skull are established within the first 3 weeks and change minimally after reaching adulthood[17]. Thus the interpolated surface from a single mouse skull served as a template to digitally design generalized transparent skulls (See-Shells) using computer-aided design (CAD) software. The frame was 3D-printed out of polymethylmethacrylate (PMMA) onto which a thin, flexible and transparent polyethylene terephthalate (PET) film was bonded (Fig. 1a and Supplementary Fig. 3). The 3D-printed frame also incorporated screw holes for fastening a custom-designed titanium head-plate for head-fixing the animal during experiments.

PET was chosen for the transparent element as this polymer has excellent optical properties[18] and is biocompatible[19]. The optical properties of the PET film were compared to the current gold standard, 170 μm thick glass coverslips (#1.5) used for a variety of microscopic imaging experiments. Sub-diffraction limit 200 nm fluorescent beads were imaged using a high magnification (×40) objective through glass coverslip and PET film to construct the lateral and axial point spread functions (PSFs, Fig. 1b, c). For beads imaged through PET, the full width at half maximum (FWHM) of the lateral PSF (Fig. 1b) was $425 \pm 26.2$ nm (mean ± standard deviation, $n = 15$) and was $409.3 \pm 13.7$ nm ($n = 5$) for beads imaged through glass coverslip. No significant difference was observed between PET and coverslip glass (Fig. 1d top, $p = 0.35$, Welch's $t$-test). For beads imaged through PET, the FWHM for the axial PSF (Fig. 1c) was $2.88 \pm 0.08$ μm ($n = 15$) and was $3.01 \pm 0.10$ μm ($n = 5$) for beads imaged through glass coverslip. Again, no significant difference was observed (Fig. 1d bottom, $p = 0.34$, Welch's $t$-test). Additional experiments characterized the light transmittance of three PET samples using a widely tunable laser at common wavelengths used for 2P imaging. All three PET samples yielded light transmittance of >90% (Supplementary Table 1). On average, across all wavelengths, light transmittance was $91.17 \pm 0.94$, $90.85 \pm 1.1$ and $91.71 \pm 0.78\%$ in the three samples tested, compared to $92.79 \pm 0.82\%$ in a standard glass coverslip.

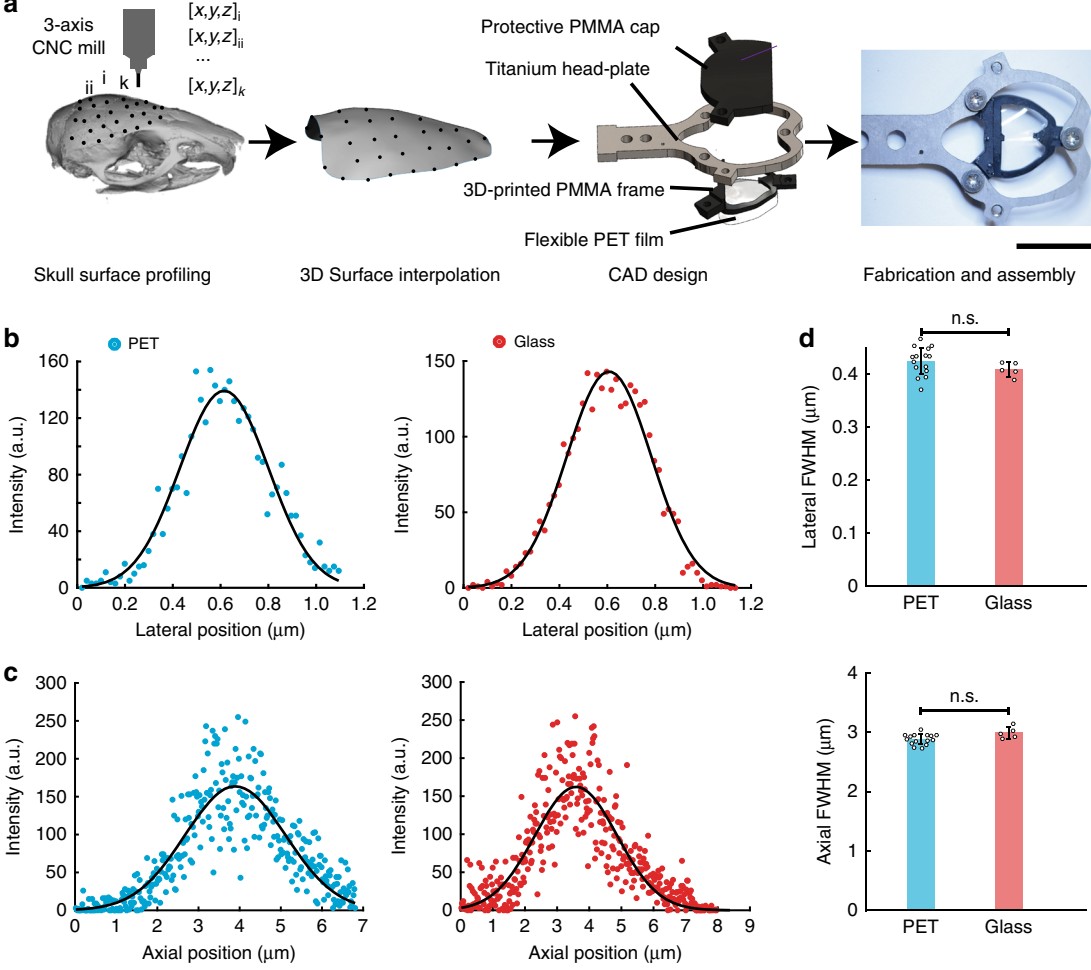

**Fig. 1** Digitally generating See-Shells. **a** The dorsal surface of the mouse skull is profiled using a computer numerical controlled mill integrated into the stereotaxic instrument. This skull surface profile is used to interpolate a three-dimensional (3D) surface. The 3D surface is used as a template to design morphologically conformant transparent implants (See-Shells), consisting of a 3D-printed polymethylmethacrylate (PMMA) frame, onto which a thin, optically clear and flexible polyethylene terephthalate (PET) film is bonded. A titanium head-plate fastened to the frame provides mechanical support for head-fixation and a 3D-printed cap protects the implant and underlying brain tissue. Photograph of a fully fabricated and assembled See-Shell is illustrated on the right. Scale bar indicates 1 cm. **b** Lateral point spread functions (PSFs) of 200 nm yellow green (YG) fluorescent beads imaged with a ×40 (1.15 NA) objective through the PET film (left) and glass coverslip (right). Black curved line indicates Gaussian fit to the intensity measurements. **c** Axial PSFs of 200 nm YG fluorescent beads imaged with the same objective as in **b**, through the PET film (left) and glass coverslip (right). Black curved line indicates Gaussian fit to the intensity measurements. **d** Bar plots overlaid with the dot plots of lateral (top) and axial (bottom) full width at half maximum (FWHM) of the PSF measured through the PET film and #1.5 glass coverslip ($n = 15$ measurements in each of three PET films, and $n = 5$ measurements in glass coverslip). a.u. arbitrary unit, n.s. not significant. Error bars indicate s.d.

Multi-photon and time-correlated single-photon counting (TCSPC) based fluorescence-lifetime imaging microscopy (FLIM) of fluorescent yellow-green (YG) beads was performed to assess whether the PET film introduces changes in light intensity or fluorescence lifetime, respectively. While the present application for See-Shells is fluorescence intensity measurements, FLIM has an additional utility due its sensitivity to changes in the tissue microenvironment and sample conditions without being affected by changes in fluorophore concentration. Concerning light intensity, 2P imaging of YG beads through the PET film required 1–2% higher gain settings on the photomultiplier tube (PMT) compared to glass coverslips. This was expected, as PET film has a slightly lower light transmission efficiency. Given that the imaging was performed well within the PMT specifications, the reduction in light transmission can be easily overcome by increased laser power or PMT gain settings or a combination of both. Next FLIM imaging was performed using the same 2P instrument. Mean fluorescence lifetime of three PET film samples

were $2.13 \pm 0.01$, $2.16 \pm 0.07$, and $2.15 \pm 0.02$ ns ($n = 3$ measurements in each substrate), comparable to the mean fluorescence lifetime of #1.5 glass coverslips ($2.15 \pm 0.01$ ns, $n = 3$ measurements, Supplementary Fig. 4). The mean lifetimes are within the resolution limit of this 2P-based FLIM system. Therefore, PET has negligible effects on both 2P and FLIM imaging.

**Chronic implantation of See-Shells.** See-Shells were chronically implanted on wild-type C57BL/6 ($n = 9$), Thy1-GCaMP6f ($n = 31$), Thy1-YFP mice ($n = 3$), and Thy1-GFPm mice ($n = 3$) after a craniotomy was performed to remove the skull over the dorsal cortex (Fig. 2a, see Methods). The median duration of the implantation was 92 days, with durations ranging from 7 to 337 days. Of these, some procedures failed because the dental cement used to seal the implants (see Methods) had not sufficiently adhered to the skull. These experiments were terminated within 10 days of surgery. Thus the overall surgery success rate was 93.5% (3 surgeons, $n = 43/46$ mice). A subset of the mice was

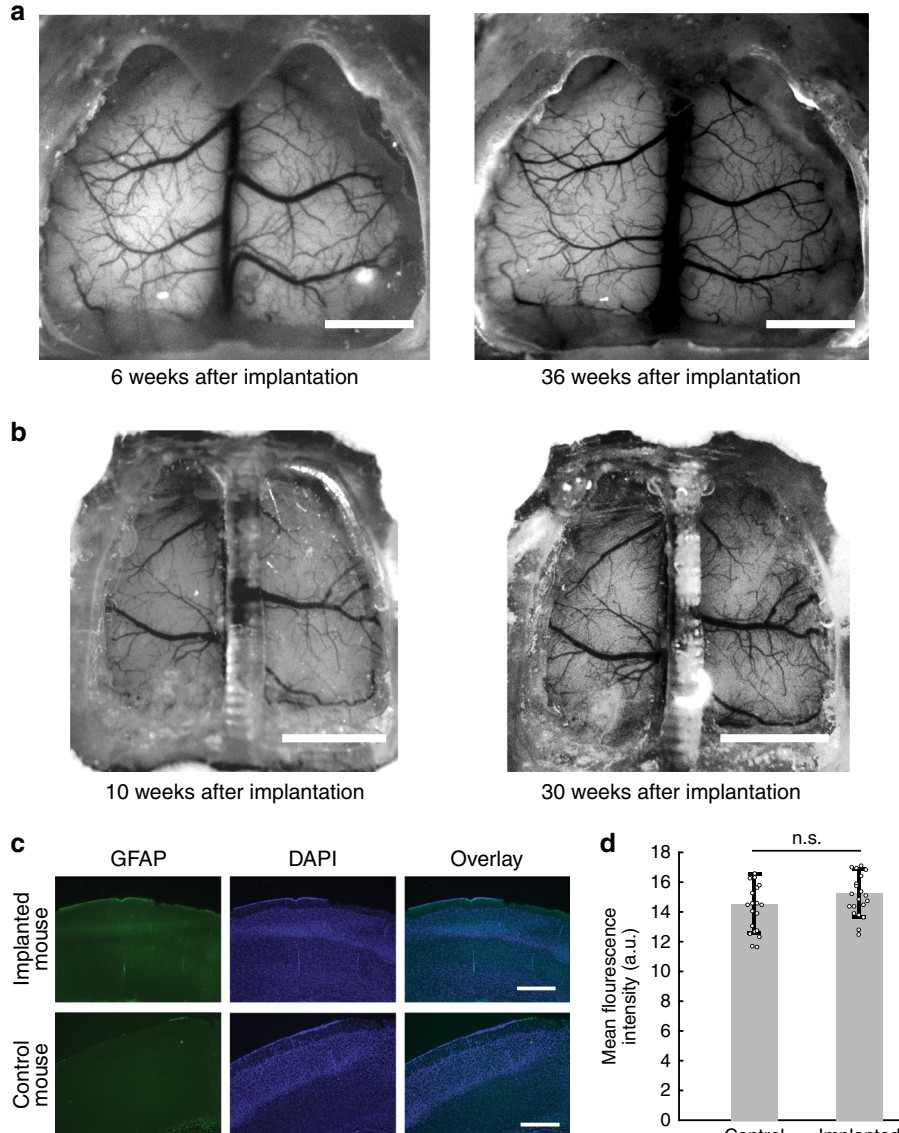

**Fig. 2** Chronic implantation of See-Shells. **a** Photographs of a Thy1-GCaMP6f mouse 6 and 36 weeks after implantation with See-Shells. Scale bars indicate 2 mm. **b** The design can be modified to fit different skull morphologies. Photographs of a *tg/tg* mouse 10 and 30 weeks after implantation. Scale bar indicates 2 mm. **c** Immunohistology analysis of mice chronically implanted with See-Shells for 5 weeks compared to age-matched non-surgical control mice at ~2.0 mm posterior to bregma and 1.5 mm lateral to the midline. Coronal slices were immunolabeled with anti-glial fibrillary acidic protein (anti-GFAP) and DAPI. No activated astrocytes were observed in any of the mice assessed. Scale bar indicates 500 μm. **d** Bar plots overlaid with the dot plots of mean GFAP fluorescence measured in 21 regions of interest (ROIs) in See-Shell implanted and non-surgical control mice. Error bars indicate s.d.

observed under a high magnification (×6) stereo-zoom microscope after implantation to assess implant opacity or bone regrowth. In 75% of the mice (*n* = 18/24), no opacity of the windows or bone regrowth was observed in any part of the field of view (FOV). In 3 mice, significant opacity (10–30% of the FOV) was observed within 60 days. In 3 mice, opacity blocking optical access to <10% of the FOV along the midline suture was observed after >100 days of implantation, with 48 weeks being the longest duration assessed.

As Thy1-GCaMP6f and Thy1-YFP mice are derived from C57BL/6 lines, implants based on the surface profile from the C57BL/6 mouse readily fit these transgenic mice. The digital design methodology used to generate the See-Shells allows easy modifications to fit skull morphologies different from commonly used wild-type mouse strains. As an example, See-Shells were custom designed for the tottering (*tg/tg*) mouse, a strain that has a mutation in the *Cacna1a* gene[20] and has a narrower skull than

C57BL/6 mice of the same age (Supplementary Fig. 2). Similar to chronic implantations on C57BL/6 and derivative mouse strains, the modified See-Shells on *tg/tg* mice (*n* = 5) remain optically clear for up to 30 weeks (Fig. 2b).

In a subset of mice (*n* = 3), the inflammatory effect of chronic implantation was assessed after 5 weeks of implantation by immunostaining for expression of glial fibrillary acidic protein (GFAP), a marker for chronic inflammation. No activated astrocytes were observed in the cortex of the implanted and control mice consistent with previous studies[21]. To assess whether there is any increased expression of GFAP, the fluorescent intensity was measured in multiple cortical areas using the methodology described previously[22]. GFAP fluorescence intensity in arbitrary units (a.u.), in the areas assessed, was 15.26 ± 1.64 a.u. in the implanted mice. In comparison, GFAP fluorescence intensity was 14.55 ± 1.99 a.u. in naive control mice (*n* = 21 measurements from 3 mice each, Student's *t*-test,

$p = 0.2971$, Fig. 2c, d). Thus the See-Shells can be implanted on mice for long durations of time and allow longitudinal imaging of the dorsal cortex.

**Sub-cellular resolution structural imaging across the cortex.** Chronically implanted cranial glass windows have been used for high-resolution imaging of neural structure in vivo over extended periods of time[11]. Several of these studies have revealed key cellular and structural mechanisms underlying experience-dependent plasticity[23–25]. Therefore, we assessed the structural imaging capability of See-Shells across the large FOV with spatial resolution and imaging depths comparable with glass cranial windows. See-Shells were implanted on Thy1-YFP mice ($n = 3$) that express the yellow fluorescent protein (YFP) in layer 2/3 and layer 5 pyramidal neurons of the cortex[26]. Figure 3a, top left

shows a mesoscale image obtained using a wide-field epi-fluorescence microscope at ×1 magnification. Multiple locations distributed across the cortex of the same mouse were then imaged at high-resolution with a 2P microscope. Whole cortical columns ($360 \times 360 \ \mu m^2$) were reconstructed up to depths of 600 μm (Fig. 3a, inset #1). Capturing multiple z-stacks from adjacent tiles allowed reconstruction of large contiguous volumes of tissue spread across millimeters of the cortex (Fig. 3a, inset #2 and #3). At an optical zoom of ×4.4 using a ×25 objective, individual neurons and their processes were imaged at depths of ~300 μm from the pial surface (Fig. 3a, inset #4). At a zoom of ×7, finer sub-cellular structures including dendrites and dendritic spines were clearly visible (Fig. 3a, inset #5 and #6). In Thy1-GFPm mice[26], the same dendrites and dendritic spines can be imaged across multiple days and imaged at multiple locations throughout

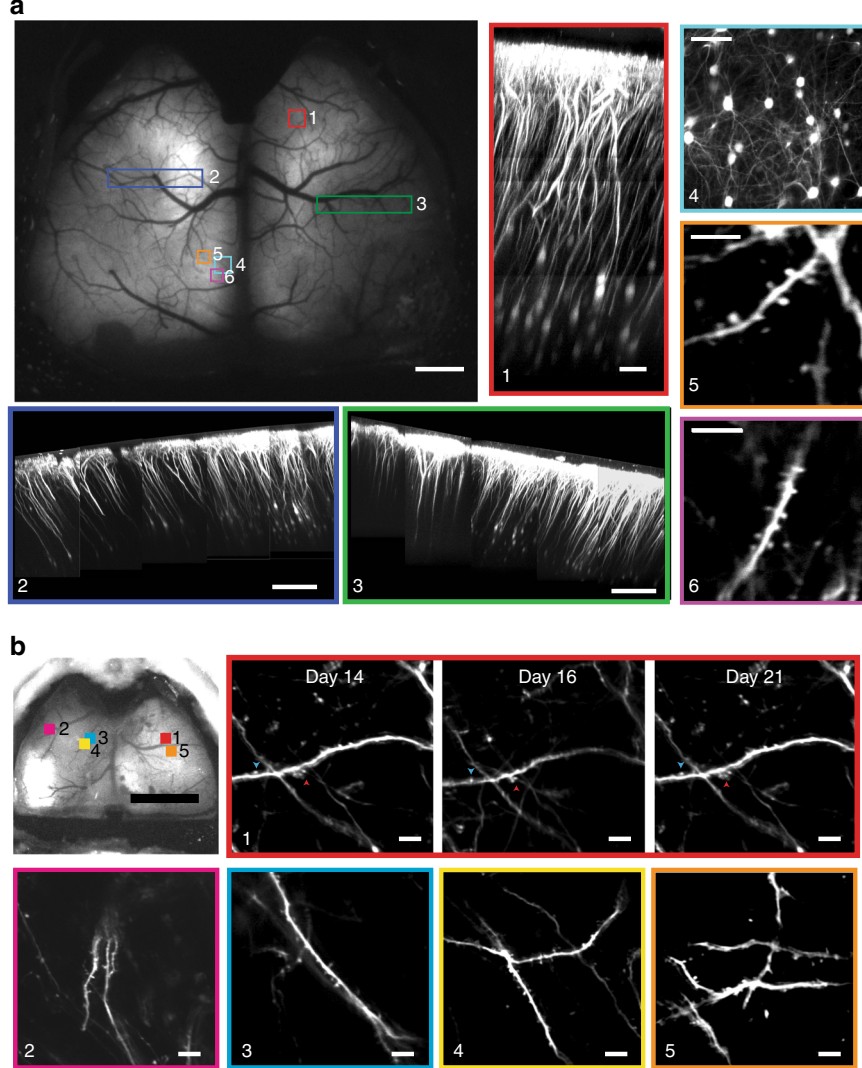

**Fig. 3** See-Shells allow structural imaging from mesoscale to microscale. **a** Wide-field image of a Thy1-YFP mouse implanted with a See-Shell taken 2 weeks after implantation. Locations marked with colored blocks were targeted for two-photon (2P) imaging. Scale bar indicates 1 mm. Inset #1: Imaging of a whole cortical column at the location marked by red block in the wide-field image. Scale bar indicates 80 μm. Inset #2 and #3: Composite images of cortical columns imaged from large contiguous areas denoted by the blue and green blocks in the wide-field image. Scale bar indicates 200 μm. Inset #4: High-resolution image of layer 2/3 pyramidal neurons imaged from the cyan block denoted in the wide-field image. Scale bar indicates 80 μm. Inset #5 and #6: Dendrites with dendritic spines of layer 2/3 neurons imaged at ~245 μm depth from the orange and magenta blocks shown in the wide-field image. Scale bars indicate 10 μm. **b** Wide-field image of a Thy1-GFPm mouse implanted with a See-Shell obtained 2 weeks after implantation. Locations marked with colored blocks were targeted for 2P imaging. Scale bar indicates 2 mm. Inset #1: Longitudinal 2P structural imaging of dendritic structures within the red block outlined on the wide-field image on the left taken in multiple experiments over 1 week. Scale bar indicates 10 μm. Insets #2–#5: Multi-site 2P imaging of dendritic structures at locations indicated by blocks 2–5 in the wide-field image. Scale bar indicates 10 μm

the cerebral cortex (Fig. 3b). Thus See-Shells allow the study of fine sub-cellular structures of neurons across the cerebral cortex.

**Ca²⁺ imaging at multiple spatial scales across the cortex.** To assess the capabilities to monitor cortical neural activity, See-Shells were implanted on Thy1-GCaMP6f mice that express the Ca²⁺ indicator GCaMP6f in layer 2/3 and layer 5 pyramidal neurons[27]. Wide-field imaging using a standard epi-fluorescence microscope captured mesoscale activity across the entire FOV (Supplementary Fig. 5, Supplementary Movie 1). Robust activation of the entire cortex was observed during locomotion, with spontaneous activity observed even at rest. In the mouse shown in Fig. 4a, four random areas highlighted by the colored blocks were targeted for 2P imaging. At each area, z-stacks of 365 × 365 μm² were captured when the mouse was fully awake and head-fixed on a custom-built disk treadmill (Fig. 4b). Two-dimensional (2D) maximum intensity projections of the z-stacks revealed macroscopic anatomical features such as blood vessels that could be matched with wide-field images to determine the imaging locations post hoc. In each tile, time series of Ca²⁺ activity were acquired in planes at 200–300 μm from the pial surface.

Individual cells were readily visualized in the average intensity projections of the time series (Fig. 4c). Spontaneous Ca²⁺ activity traces from a small subset of the detected neurons in each tile during awake head-fixation are shown in Fig. 4d.

To evaluate See-Shells' capability to monitor Ca²⁺ signals in the same neurons over time, multiple imaging sessions were performed in the same FOV over a month starting 44 weeks after implantation on a Thy1-GCaMP6f mouse (Fig. 4e). Average intensity projections from a set of high-resolution images qualitatively indicated that the same neurons could be identified over time. Robust Ca²⁺ signals were obtained with the peak ΔF/F of randomly selected individual neurons ranging between 128.7 to 240.4% across all imaging sessions. Linear regression analysis of peak ΔF/F measured over imaging sessions indicated that Ca²⁺ signals were diminished slightly across duration evaluated ($R^2 = 0.378$, average slope of trend-line $= -1.23\%$ per day, $n = 5$ neurons, Supplementary Fig. 6). Further, 2P imaging was performed at multiple sites distributed across the cortex in a late-stage implanted mouse (Day 335, Supplementary Fig. 7). Similar to the data shown in Fig. 4, robust Ca²⁺ signals were acquired from each region with peak ΔF/F of individual neurons ranging between 61.51 to 191.84%.

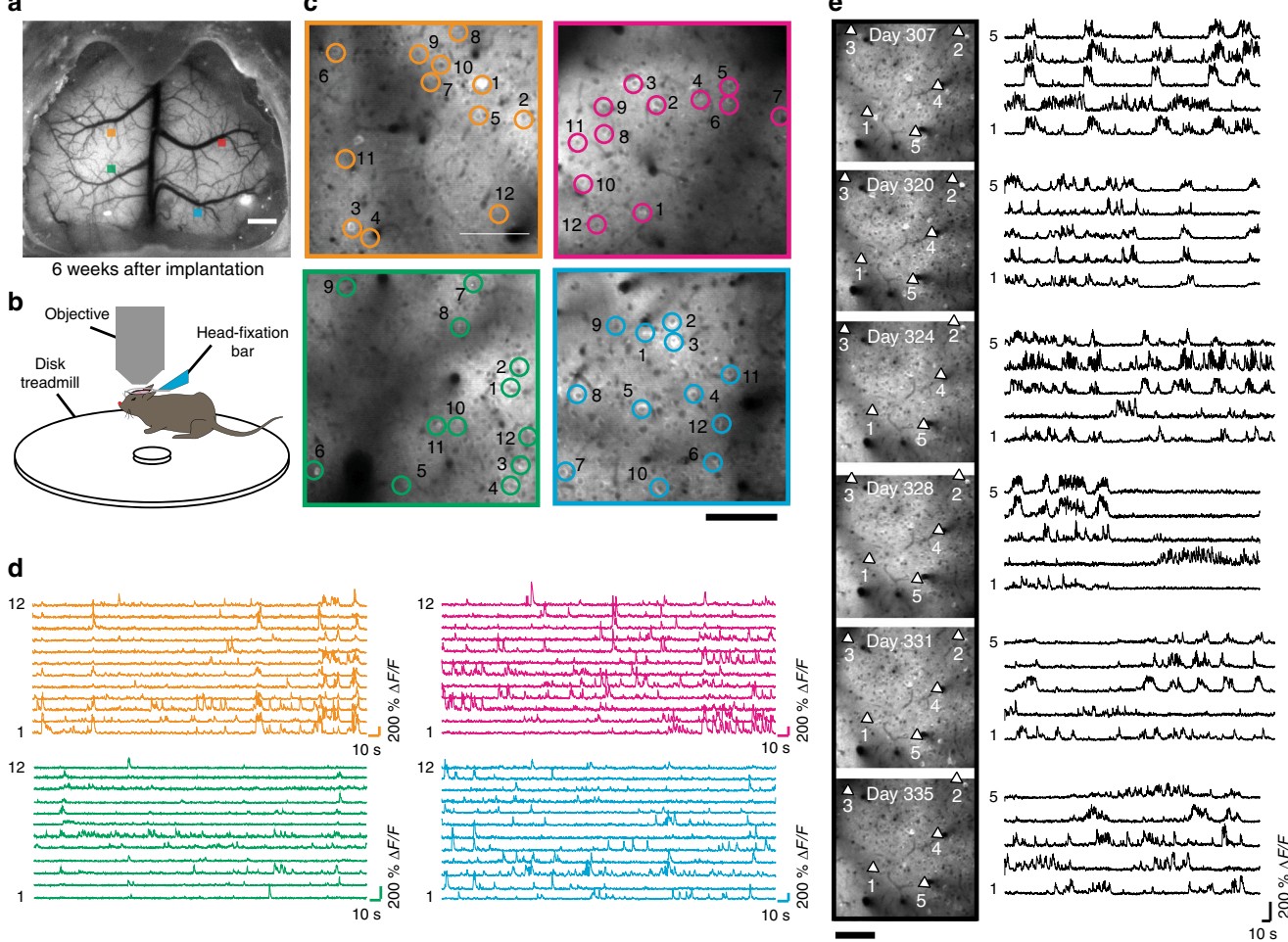

**Fig. 4** Monitoring Ca²⁺ activities in awake head-fixed mice. **a** Wide-field image of a Thy1-GCaMP6f mouse implanted with a See-Shell taken 6 weeks after implantation. Four locations indicated by the colored blocks were imaged using a 2P microscope. Scale bar indicates 1 mm. **b** Schematic of the mouse on the custom-designed disk treadmill used for 2P imaging. **c** Average intensity images calculated from 5-min time series acquired 200–300 μm deep from pial surface. Scale bar indicates 100 μm. **d** Color-coded time series of Ca²⁺ activities of neurons identified and annotated by open circles in the respective average intensity images in **c**. **e** Time lapse Ca²⁺ imaging: Left: Average intensity images calculated from 5 min time series acquired from the same field of view in a See-Shell-implanted Thy1-GCaMP6f mouse in six separate imaging sessions. Activities of individual neurons, indicated by arrows, are shown in the right. Scale bar indicates 100 μm

Further, $Ca^{2+}$ activities were tracked in Thy1-GCaMP6f mice using 2P imaging in the hindlimb area of the primary motor cortex along with high-speed video monitoring. Mice implanted with See-Shells readily performed a variety of behaviors including walking and grooming during head-fixation. In the recording highlighted in Supplementary Movie 2, increased $Ca^{2+}$ activity occurred during walking as tracked by movements of the hindlimb, forelimb, and disk treadmill but the modulation was absent during grooming, indicated by forelimb movement (Supplementary Fig. 8).

Thus See-Shells allow multi-scale imaging over long durations in the mouse cerebral cortex during a wide range of head-fixed behaviors. The capability to image in the same animal structurally at subcellular resolution as well as $Ca^{2+}$ activity at the cell and mesoscale level will provide new insights between factors, such as physical structure and neural state.

**Multi-modal and bidirectional neural interfacing.** Another advantage of the See-Shell design and PET film is incorporation of additional modalities to record and/or perturb neural structures. For example, combining wide-field $Ca^{2+}$ activity monitoring while simultaneously recording neural firing from localized circuits will enable determination of how global cortical activity relates to local circuit activity[15]. See-Shells were engineered with a ~1.5 mm perforation over the primary somato-sensory cortex to introduce a 32-channel, silicon-based recording probe (Fig. 5a). The perforations were made prior to implantation and sealed with quick setting silicone sealant that could be removed during experiments ($n = 3$ Thy1-GCaMP6f, Fig. 5b).

Mesoscale $Ca^{2+}$ imaging was performed simultaneously with the single cell recordings in left primary somatosensory cortex and high-speed behavioral recording during awake head-fixation (Fig. 5c). $Ca^{2+}$ activity from six regions of interest (ROIs) in the bilateral motor (M1), somatosensory (S1), and visual cortices (V1) show robust co-activation of multiple homotopic regions. Single-cell electrophysiology recordings revealed a subset of neurons with increased firing rates correlated with mesoscale $Ca^{2+}$ activity in the ipsilateral primary somatosensory cortex (I-S1) (Fig. 5c, d). To assess the relation between the $Ca^{2+}$ activity and activity of individual neurons, the $Ca^{2+}$ signal in I-S1 was cross-correlated with the spike firing rate of each recorded neuron and compared

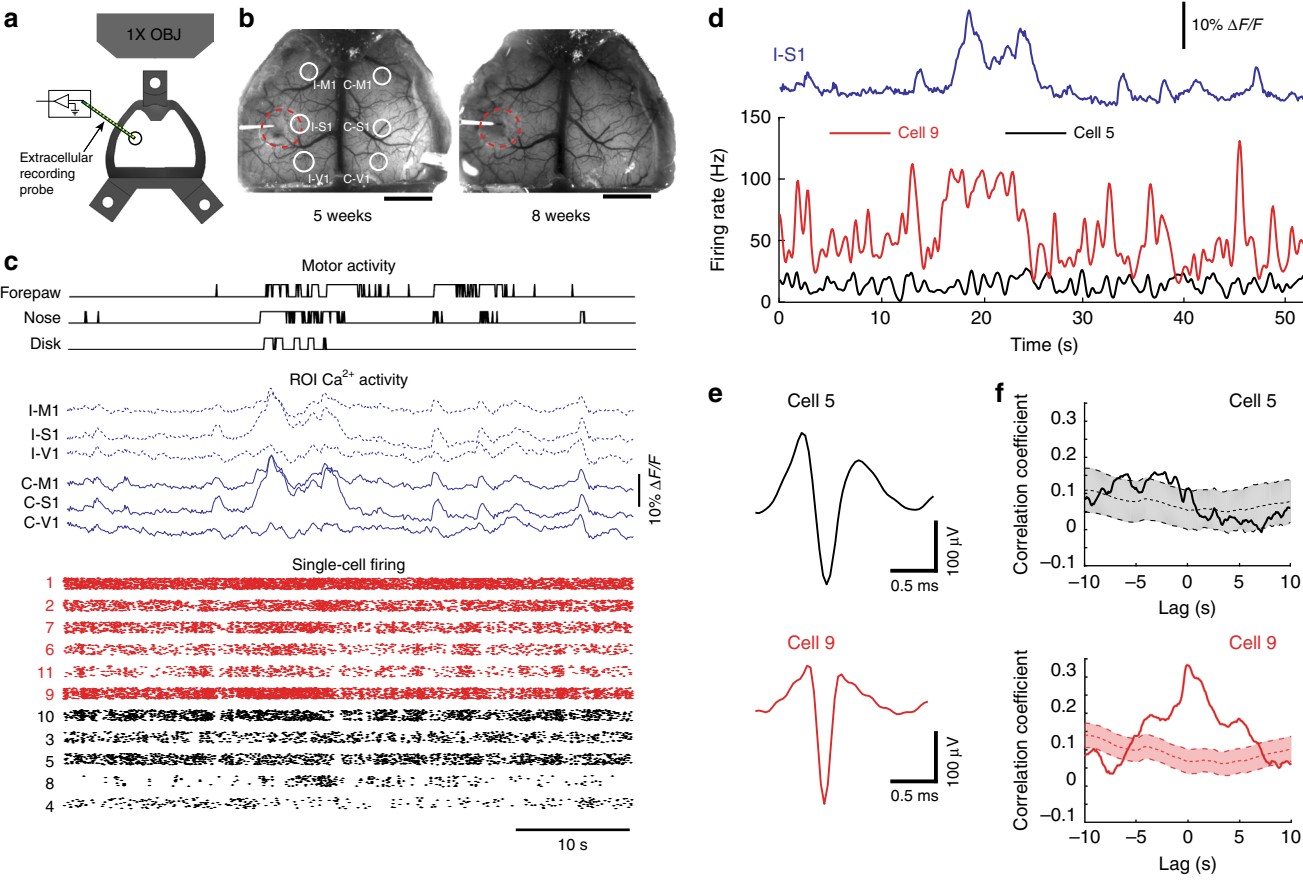

**Fig. 5** Simultaneous extracellular recordings with wide-field $Ca^{2+}$ imaging and behavioral tracking. **a** Schematic of implanted See-Shells with perforation over the primary somatosensory cortex allowing insertion of a multi-channel silicon-based neural probe. **b** Photographs of a Thy1-GCaMP6f mouse implanted with a perforated See-Shell taken during two experimental sessions. Red dashed lines indicate the outline of perforation. White circles indicate regions of interest (ROIs) analyzed for extracting $Ca^{2+}$ traces. I-M1 ipsilateral primary motor cortex, I-S1 ipsilateral primary somatosensory cortex, I-V1 ipsilateral primary visual cortex, C-M1 contralateral primary motor cortex, C-S1 contralateral primary somatosensory cortex, C-V1 contralateral primary visual cortex. Scale bars indicate 2 mm. **c** Simultaneously recorded disk, nose, and forelimb movements, aligned with $Ca^{2+}$ activity traces in the ROIs indicated in **b**, and single unit spike raster plots recorded from the multi-channel silicon-based neural probe. Red raster plots indicate neurons with spike firing rates correlated with $Ca^{2+}$ activity in I-S1. Individual points from each cell were slightly shifted in a randomized fashion in the *y* axis for ease of visualization. **d** Spike firing rates of two representative cells, with one that was correlated with $Ca^{2+}$ activity in I-S1 (cell 9) and one that did not show correlation (cell 5). **e** Average action potential waveforms of the two cells shown in **d**. **f** Cross-correlation of firing rates of representative cells in **d** and $Ca^{2+}$ activity in I-S1 with 95% confidence interval of cross-correlations with 1000 bootstrapped shuffled trials of the firing rate to determine significance

to the 1000 randomly shuffled spike firing rate of each cell (See Methods)[28]. As shown for two representative cells, one has spike firing rate correlated with the $Ca^{2+}$ signals in I-S1 (cell 9) and one that did not (cell 5, Fig. 5e, f). At zero lag, 6 of the 11 neurons had significant correlation with the $Ca^{2+}$ signals in I-S1 (correlation coefficient > mean + 1.98 s.d. of the shuffled traces). These experiments suggest that activities of individual neurons are diverse in terms of their correlation with mesoscale activity. See-Shells thus allow us to observe cortical activity at multiple scales and understand their significance to behavior.

Finally, we demonstrate that perforations in See-Shells introduced after chronic implantation can be used to perturb neural circuits with intracortical microstimulation, which has been widely used to assess cortical connectivity and function including effects on downstream targets. In a subset of Thy1-GCaMP6f mice ($n = 3$, Fig. 6a), the PET film was carefully punctured with a sterile syringe needle to introduce microstimulation electrodes. Stimulation resulted in robust activation of both hemispheres in the awake and anesthetized states (Fig. 6b, c) with significantly prolonged responses in the anesthetized mouse compared to responses in the mouse when awake. Traces shown in Fig. 6c represent the average trace of ≥10 trials for the awake state, and ≥4 trials for anesthetized state in each mouse (Mouse #1: 10 awake, 5 anesthetized, Mouse #2: 15 awake, 6 anesthetized, and Mouse #3: 11 awake, 4 anesthetized). The FWHM of the post-stimulus $Ca^{2+}$ fluorescence response was significantly longer in all ROIs examined during isoflurane anesthesia (Fig. 6d, Student's $t$-test, $p < 0.001$). These results with See-Shells in the awake animal

extend previous flavoprotein imaging observations in anesthetized mice showing that microstimulation of the primary motor cortex co-activates homotopic regions via the corpus callosum[29]. Therefore, See-Shells can be used to study the effects of perturbing localized regions and suggest that the arousal state alters cortical dynamics. While the present study employed intracortical microstimulation, the methodology could be easily compatible with optogenetic or chemical stimulation of cortical or sub-cortical brain regions.

## Discussion

We developed See-Shells, transparent, morphologically conformant polymer skulls that allow optical access to a large part of the dorsal cerebral cortex for high-resolution structural and functional imaging. These windows can be implanted for long periods and remain functional for over 300 days. In line with estimates from recent studies using curved glass windows[14], See-Shells provide optical access to ~1 million neurons from the cortical surface. In addition, optical imaging with See-Shells can be combined with other modalities. Perforation of the PET film allows access to the brain underneath the implant and here we demonstrated wide-field $Ca^{2+}$ imaging with simultaneous intra-cortical microstimulation and electrophysiological recordings.

The optical properties of PET compare favorably with glass coverslips when evaluated by 2P or FLIM imaging. The latter opens up the possibility for FLIM intra-vital brain imaging of auto-fluorescence using PET windows[30,31]. For example, the

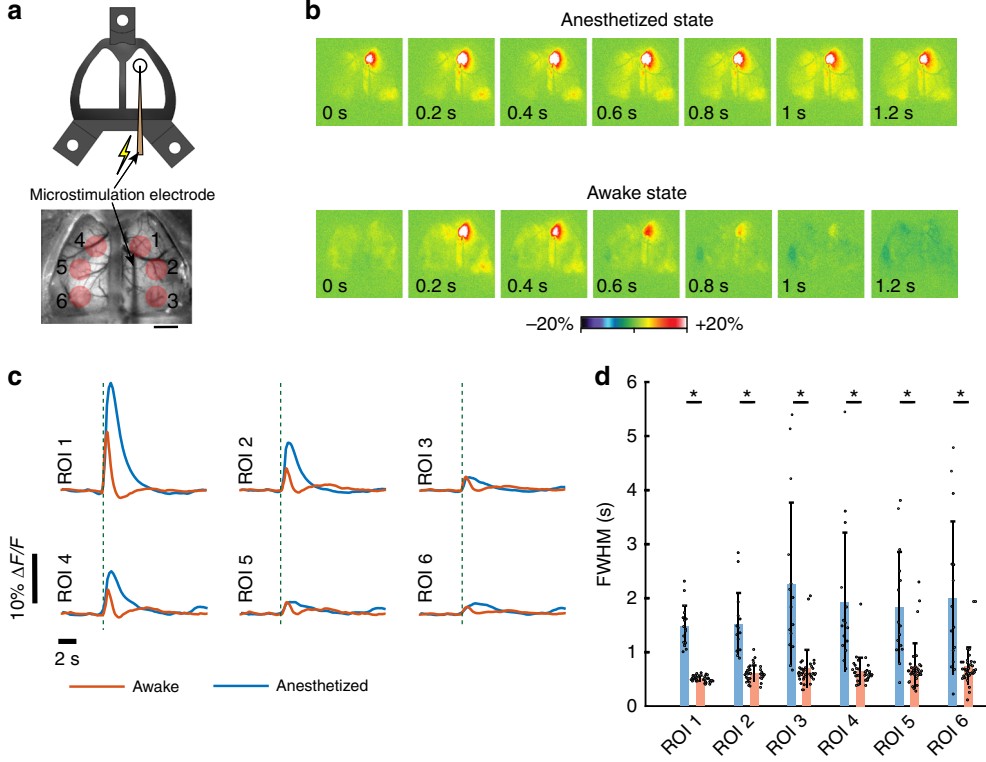

**Fig. 6** Cortical microstimulation during wide-field $Ca^{2+}$ imaging. **a** Top: Cartoon schematic of a microstimulation electrode inserted through a perforated See-Shell. Bottom: a wide-field image showing microstimulation electrode inserted at the primary motor cortex. Red circles indicate regions of interest (ROIs) analyzed. Scale bar indicates 2 mm. **b** Pseudocolor plots of normalized change in fluorescent intensity in response to primary motor cortex microstimulation. Top: under isoflurane anesthesia, bottom: awake. Post-stimulation time stamps are indicated at the bottom of each image. **c** Average normalized $Ca^{2+}$ activity traces in response to microstimulation of primary motor cortex for different ROIs indicated in **a**. ROI 1: primary motor cortex and stimulation site; ROI 2: ipsilateral somatosensory cortex; ROI 3: ipsilateral visual cortex; ROI 4: contralateral motor cortex; ROI 5: contralateral somatosensory cortex; and ROI 6: contralateral visual cortex. Dashed lines indicate the time of stimulus. **d** Bar plots overlaid with the dot plots of full width at half maximum in the different ROIs under anesthetized and awake conditions. Error bars indicate s.d.; Student's $t$-test, *indicates $p < 0.001$

intrinsically fluorescent metabolites nicotinamide adenine dinucleotide hydrogen and flavin adenine dinucleotide are widely used in vivo to record label-free cellular activity based on their oxidation state[32–34]. Changes in the lifetime of both coenzymes are used to monitor the biological microenvironment[32], including in intra-vital studies[35]. Thus, while the current goal is to use PET to realize transparent skulls for cortex-wide imaging, the flexibility, optical clarity, and biocompatibility demonstrate the feasibility of engineering anatomically realistic windows for intra-vital imaging in a wide variety of organs, such as mammary gland[36] and lung[37].

See-Shells can be implanted using simple modifications to well-established chronic cranial window implantation procedures[38,39], with the major change being the removal of large sections of the skull above the dorsal cerebral cortex. In this study, we utilized a robot that uses surface profiling to guide a computer numerical controlled (CNC) mill to perform the craniotomy[40]. Automation enabled reliable removal of the bone without damage to the underlying dura and brain and also allowed precise positioning of the implant relative to bregma. We also performed manual craniotomies for See-Shells implantations on *tg/tg* mice, demonstrating that automation of the craniotomy is not a pre-requisite for successful implantation, although it could help investigators quickly adapt these tools for their research.

High-quality mesoscopic $Ca^{2+}$ imaging in the awake animal has been performed in mice implanted with See-Shells for 48 weeks, the longest period tested to date. Chronic imaging over this duration provides the opportunity for very long-term studies of brain development, plasticity and learning, disease processes, and evaluation of new therapies. For developmental studies, the primary limitation will be skull growth in the postnatal period. Mouse skull sutures that fuse do so by ~45 days of age, and most cranial expansion is complete by 6 weeks[17,41]. However, the majority of skull growth is complete earlier (~2–3 weeks of age), suggesting that the windows could be implanted in younger animals.

Several aspects of the design and fabrication of the See-Shells are widely adoptable and highly flexible. See-Shells can be fabricated using desktop tools and are inexpensive (<$20 each). Once the individual components are fabricated (or procured from commercial fabrication services), the implant can be assembled in <15 min. Therefore, this is a tool that can be readily adopted by most laboratories. Although the cranial implants were developed for the dorsal cerebral cortex with its fairly regular convex surface, the design can be modified for a variety of skull morphologies. Future versions can be designed to cover not only the dorsal cerebral cortex but also other regions including the olfactory bulb, cerebellum, and more lateral cortical regions such as the auditory cortex.

See-Shells could also be engineered for optical interfacing with complex and mobile anatomical structures such as the spine. The 3D-printed frame can be modified to incorporate mounting features to precisely attach miniaturized microscopes[42] and wirelessly controlled devices for infusing pharmacological agents or performing optogenetic stimulations[43]. Recently, ultra-miniaturized lens-less fluorescence microscopes <1 mm in size have been developed[44]. Engineering See-Shells embedded with these miniaturized lens-less imaging systems offers the possibility of monitoring the activity of the whole cortex during freely moving behaviors.

The ability to simultaneously monitor local microcircuit activity using extracellular recordings combined with wide-field $Ca^{2+}$ imaging offers the potential to integrate the contribution of local microcircuits to mesoscale activities. We show that both mesoscale and single unit activities correlated with motor activity. Additional information about the cortical state is likely represented in the complex spatio-temporal patterns of activity

observed with mesoscale imaging. See-Shells would allow systematic multi-scale studies tying activity at microcircuits to large scale network activity. Using See-Shells to combine wide-field $Ca^{2+}$ imaging with in vivo patch clamping methodologies to record from single[45] and multiple neurons[46] will help us to better understand how mesoscale network activity influences subthreshold membrane potential dynamics in individual neurons.

We have used perforated See-Shells to perform intracortical microstimulation. In addition to stimulating electrodes, this methodology can be used to introduce probes for optogenetic or chemical perturbation of cortical and sub-cortical brain regions. These perturbation strategies will be particularly useful to study how activation or inhibition of localized brain regions or circuits affect global cerebral cortical activity. This includes both specific pathways such as the cerebello-thalamo-cortical[47,48] and basal ganglia-thalamo-cortical projections[49,50] or the more diffuse neuromodulatory projections such as noradrenergic inputs from the brain stem[51,52] or cholinergic inputs from the basal forebrain[53].

With their similar genetics, anatomy, physiology, and behavioral repertoire, non-human primates (NHPs) provide the closest animal model to humans for understanding both normal functions as well as disease[54–56]. Emerging genetic modification techniques, including CRISPR, make generating transgenic NHPs with broad expression of $Ca^{2+}$ reporters a possibility[57]. Further, techniques for long-term 2P imaging in NHPs are also emerging[58]. The See-Shells methodology can be adapted to build customized implants derived from computed-tomographic (CT) scans of the skull, enabling imaging of neural activities across centimeters of the NHP cortex.

While the See-Shells allow sub-cellular resolution imaging across the cortex, imaging across the whole FOV simultaneously at high resolution is currently not possible. Recently developed wide-field 2P imaging systems[8–10] allow simultaneous imaging of nearly an entire hemisphere. Extending such optical systems to simultaneously image the whole dorsal cortex at cellular resolution would be very powerful. We have performed mesoscale imaging while simultaneously inserting neural probes for extracellular recordings and microstimulation. It is difficult to do such experiments with 2P microscopes, given the short working distance of high NA and high-magnification objectives. In the future, it may be possible to perform random access 2P imaging across the whole FOV using customized long working distance high-resolution objectives or by using optical relays between the PET surface and the objectives to allow more room for the introduction of recording probes[59]. Neural probes that are specifically designed to be compatible with 2P imaging could also be engineered, or flexible neural probes that can be reconfigured after implantation to provide unimpeded optical access could be used[60,61].

## Methods

**See-Shell design and fabrication.** All of the animal studies were approved by and conducted in conformity with the Institutional Animal Care and Use Committee of the University of Minnesota. An adult C57BL/6 male mouse (8 weeks, #000664, Jackson Laboratories) and a *tg/tg* male mouse (16 weeks) were used for skull surface profiling. In each experiment, the mouse was anesthetized using isoflurane in oxygen (4–5% induction, 0.8–1.5% maintenance). The scalp was shaved and sterilized using standard aseptic procedures, after which the mouse was head-fixed in a stereotaxic instrument (David Kopf Instruments Inc.) which was modified to have CNC milling capabilities by incorporating a programmatically controlled three-axis motorized manipulator (MTS25-Z8, Thorlabs)[40]. A handheld mill (Rampower, Ram Products Inc.) fitted with a 200-μm diameter end mill (Harvey tools Inc.) was mounted on the three-axis manipulator using a custom adaptor plate. A custom computer program was written in LabVIEW (National Instruments Inc.) to control the position of the manipulator.

The scalp covering the dorsal skull surface was excised and fascia removed using a micro-curette to prepare the skull for surface profiling. The end mill mounted on a motorized stage was carefully lowered until the end mill tip made contact with

the skull surface at bregma. This process was visualized at the highest magnification setting of the stereo-zoom microscope (×6, M60, Leica) to ensure that the tip did not exert a force large enough to deform the skull surface before registering the coordinates in the LabVIEW program. This served as the origin of a Cartesian coordinate system. The LabVIEW program then raised the end mill 0.5 mm above bregma and moved it laterally to the first profiling point (Supplementary Figs. 1a and 2a). The end mill was carefully lowered until it made contact with the skull surface and the z-coordinate was registered. The program then raised the end mill by 0.5 mm and moved it laterally to the next profiling point. The process of registering the z-coordinate was repeated at 85 profiling points on the dorsal skull surface of the C57BL/6 mouse and 134 points on the tg/tg mouse. These data were used to construct 3D point clouds to define the skull surfaces (Supplementary Figs. 1b and 2b).

The point cloud was imported into a CAD software (Solidworks, Dassault Systèmes). Points along the medial–lateral direction were used to define 3D curves and to interpolate a 3D surface mimicking the skull surface (Supplementary Figs. 1c and 2c). This 3D surface was then extruded to 0.6–0.8 mm thickness to create a solid surface, which was then used as a template for defining the structural frame of the See-Shell (Fig. 1a, Supplementary Figs. 1d and 2d). The CAD files used for 3D printing of the frame are available for download (Supplementary Data 1).

See-Shells were fabricated in a multistep process illustrated in detail in Supplementary Fig. 3. First, the See-Shell structural frame and two molds to assist with bonding the PET to the See-Shell frame were 3D-printed out of PMMA (RS-F2-GPBK-04, RS-F2-GPCL-04, Formlabs Inc.) using a desktop stereolithography (SLA) printer (Form 2, Formlabs Inc., Supplementary Fig. 3a, b). The three holes in the frame were tapped using a 0–80 hand tap (# 15J611, Grainger). A desktop laser jet printer (HP210w, Hewlett Packard Inc.) was used to print an outline matching the See-Shell frame on a 50 μm thick PET film (MELINEX 462, Dupont Inc.). The PET film was cleaned using ethanol and low-lint cleaning tissue (KimWipes, Kimtech Inc.). A pair of scissors was used to cut the PET film using the printed outline as a reference. This PET film was then aligned to the PMMA frame and bonded using a clear, two-part quick setting epoxy adhesive (Scotch-Weld™ DP100 Plus Clear, 3 M Inc.) (Supplementary Fig. 3c–f). A head-plate was fabricated from a 0.016 inch sheet of titanium using a water jet cutter (Omax Inc., see Supplementary Data 1 for CAD drawing). The titanium head-plate was designed such that a ×25 objective, with 3 mm long working distance could access the whole FOV provided by the See-Shells while also providing adequate mechanical support for the head-fixed experiments.

**PET optical characterization.** An inverted 2P laser scanning microscope was used to image 200 nm YG beads (Polysciences Inc., Warrington, PA, cat#18142-2) using a ×40 magnification, 1.15 numerical aperture (NA) objective (Nikon Apo LWD) through three PET film samples and compared to imaging through a #1.5 glass bottom dish. A Gaussian curve was fit using a standard curve fitting toolbox (MATLAB, Mathworks Inc.). The Gaussian profile was used to estimate the PSFs as illustrated in Fig. 1b, c. YG beads were imaged at 500 sections over 13 μm along the z-axis to construct the axial PSF (Fig. 1c). In these characterizations, identical gain settings and analyses were used for the PET and glass coverslip measurements.

Time domain FLIM was performed using TCSPC with Becker and Hickl SPC-150 board to determine the fluorescent lifetime of YG beads imaged through PET. An 80 MHz Ti:Sapphire laser (Spectra Physics; Maitai) tuned to the wavelength of 890 nm was used as the excitation source. The excitation and emission were coupled through an inverted microscope (Nikon; Eclipse TE300) with a ×20 air immersion objective (Nikon, Plan Fluor, N.A. 0.75). A 520/30 nm band-pass emission filter (Semrock, Rochester NY) was also used to selectively collect YG beads fluorescence. FLIM images were collected at 256 × 256 pixel resolution with 30 s acquisition using SPC-830 Photon Counting Electronics (Becker & Hickl GbmH, Berlin, Germany) and Hamamatsu H742MP-40 GaAsP photomultiplier tube (Hamamatsu Photonics, Bridgewater, NJ). To compare light intensity attenuation of PET with glass, the laser power was kept at a fixed value using the calibrated power control on a custom-built 2P microscope and the gain on the PMT was recorded to reach saturation. Urea crystals were used to determine the Instrumentation Response Function with a 445/20 bandpass emission filter (Semrock, Rochester, NY). SPCImage software (Becker & Hickl GbmH, Berlin, Germany) was used to analyze the fluorescence-lifetime decay curves. The lifetime decay of each pixel was fit with a single exponential decay that resulted in a $\chi^2$ error of $1.05 \pm 0.12$ ($n = 3$). Image analyses to estimate FWHM of fluorescent intensities were performed in Fiji[62].

**In vivo surgical implantation.** The procedure for implantation of the See-Shells was adapted from previously reported chronic glass window implantation protocols[39]. Mice were anesthetized using 1–3% isoflurane in pure oxygen (0.6 mL min⁻¹). The scalp was shaved and cleaned using standard aseptic surgical procedures. Eyes were covered with ophthalmic eye ointment (Puralube, Dechra Veterinary Products). Buprenorphine (0.1 mg kg⁻¹) and Meloxicam (1–2 mg kg⁻¹) were administered subcutaneously for analgesia and managing inflammation, respectively. Mice were then head-fixed using ear bars and a nose cone in the stereotaxic instrument equipped with the CNC milling machine. A feedback regulated heating pad was used to maintain the body temperature at 37 °C throughout the procedure. Anesthetic depth was assessed every 15 min via toe pinch stimulus and anesthesia

dosage adjusted as needed. Warm lactated Ringer's solution was administered subcutaneously every 2 h to prevent dehydration. A local anesthetic (1% Lidocaine) was administered subcutaneously at the incision sites. The scalp was then removed to expose the skull covering the dorsal and cerebellar cortices. A micro curette was used to scrape the skull surface to remove fascia.

The CNC milling machine[40] incorporated in the stereotaxic instrument was used to perform automated craniotomies on C57BL/6, Thy1-YFP (#003709, Jackson Laboratories), Thy1-GFPm (#007788, Jackson Laboratories), and Thy1-GCaMP6f mice (#024276, Jackson Laboratories) (Supplementary Fig. 1a). Briefly, the CNC end mill was lowered to the skull surface at pilot points along a predefined path slightly smaller than the perimeter of the See-Shell. Once the contact was confirmed visually through a stereomicroscope, the z-coordinate at that point was registered. This process was repeated at multiple points along the desired craniotomy path. The registered coordinates were then used to interpolate a 3D cutting path for milling the skull in the LabVIEW program. For each craniotomy, the initial milling depth was 50 μm, which was well within the thickness of the skull. In each subsequent milling pass, the depth was incremented in 10 μm steps until reaching the soft part of the bone or trabeculae in a section of the craniotomy path. This was sufficient to pry open the bone for excision across the whole craniotomy. Of the surgeries assessed, the milling was stopped at depths of $107.85 \pm 16.79$ μm in C57BL/6 ($n = 7$ mice), $57.14 \pm 16.25$ μm in Thy1-GCaMP6f ($n = 14$ mice), and 70 μm in three Thy1-YFP mice. Craniotomies in tg/tg mice were performed manually.

Prior to removing the milled skull, one or two self-tapping screws (F000CE094, Morris Precision Screws and Parts) were implanted 2–3 mm posterior and 3 mm mediolateral to lambda to assist in anchoring the See-Shell to the skull. The skull was removed using surgical forceps taking care to ensure the dura was intact over the entire exposed brain. The brain was covered with sterilized surgical gauze pad soaked in 0.9% saline. The See-Shell was sterilized by soaking in 70% ethanol for 2–3 min followed by rinsing in sterile saline. The gauze pad was removed and the See-Shell was gently placed on top of the exposed brain and aligned to the craniotomy (Supplementary Fig. 3g). The edge of the See-Shell was attached to the skull around the craniotomy by applying a few drops of cyanoacrylate glue (VetBond, 3 M Inc.) using a 29-gauge syringe needle. Dental cement (S380, C&B Metabond, Parkell Inc.) was applied to the periphery of the See-Shell to cement it to the skull surface. Care was taken to ensure the screw holes for fastening the titanium head-plates were not filled in with the uncured dental cement. The dental cement was allowed to fully cure. The titanium head-plate was attached to the frame using 3/32 inch flat head 0–80 screws on the day of implantation. This was followed by a second round of dental cement application to ensure that three fastening locations were fully enclosed in the cement to make it a structurally rigid implant. To protect the implant and underlying brain from light and physical impacts, an opaque 3D-printed PMMA cap was fastened to the titanium head-plate using 3/16 inch flat head 0–80 screws.

In a subset of Thy1-GCaMP6f mice implanted ($n = 3$), the See-Shells had ~1.5 mm diameter perforations above the primary somatosensory cortex (centered −0.76 mm, −2.47 mm AL to bregma). The perforation was sealed using quick setting silicone sealant (KWIK-SIL, World Precision Instruments) on the day of surgery. The brain could be accessed in multiple experimental sessions across weeks by removing and replacing the silicone seal.

After implantation, mice were allowed to recover on a heating pad until ambulatory and then returned to a clean home cage. All mice were administered Buprenorphine and Meloxicam post-operatively on the day of the surgery as well as the 3 succeeding days to assist with full recovery.

**2P imaging.** All mice were allowed to recover from surgery for ≥7 days before imaging experiments were attempted. A 2P microscope (Leica SP5II) with a ×25 (0.95 NA) water immersion objective was used for high-resolution imaging experiments in vivo. A Mai: Tai Deepsee (Spectra-Physics) laser was tuned to 940 nm wavelength for excitation. Mice were head-fixed under the 2P microscope in a custom-designed disk treadmill (Fig. 4b). Locations in the FOV were targeted at random locations as illustrated in Figs. 3a and 4a.

For structural imaging in Thy1-YFP mice, each z-stack had a FOV of 365 × 365 μm² (512 × 512 pixels), starting ~100 μm above the top surface of the PET film, with images acquired every 2 μm down to 800 μm below the starting plane. In two instances, z-stacks were acquired from 5 adjacent tiles by moving the objective in the medial–lateral direction by 340 μm such that one edge has an overlap of 25 μm.

Ca²⁺ imaging was performed in Thy1-GCaMP6f using the same head-fixation set-up in fully awake mice. Z-stacks were acquired every 10 μm from adjacent 365 × 365 μm² tiles with a 15 μm overlap along one edge. Maximum intensity projects of these z-stacks were constructed and macroscopic feature in these projects were used to determine their coordinates in the corresponding wide-field epifluorescence image. Time series were acquired at 20 Hz (256 × 256 pixels) for 5 min at one plane in each tile at depths ranging between 200 and 300 μm.

**Intracortical microstimulation during wide-field imaging.** Mice were head-fixed in the custom-designed disk treadmill placed under a stereo-zoom microscope under light (0.5–1%) isoflurane anesthesia. A feedback regulated heating pad was used to regulate the body temperature at 37 °C. The PET film was carefully per-forated (+1 mm, +1 mm, AL to bregma) using a 29-gauge needle for introduction

of an intracortical stimulation electrode. The treadmill was then placed under an epifluorescence microscope (QUANTEM: 5125 C, Nikon). A 250-μm diameter tungsten micro-electrode (Lot # 217037, FHC) was introduced into the brain at an angle of 45° (anterior–posterior direction) using a micromanipulator. Imaging was performed using a ×1 objective when the animal was anesthetized (0.5–1% isoflurane). Each trial lasted a total duration of 2 min sampled at 20 Hz using Metamorph (Molecular Devices Inc.). Stimulation train of 20 pulses (200 μA at 100 Hz) was delivered to the primary motor cortex ~5 s after initiation of each trial. Anesthesia was turned off and mouse was allowed to recover for 1 h before recording during awake state. Behavior was recorded during awake trials using a high-speed camera (FL3-U3-13Y3M-C, FLIR Inc.) at 20 frames per second to monitor whisker or limb movements. At the end of the experiment, the perforations were covered with silicone sealant and the animals were returned to the home cage.

**Simultaneous extracellular recordings and wide-field imaging**. Simultaneous extracellular recording and wide-field imaging were performed on Thy1-GCaMP6f mice that had perforations created in the See-Shells prior to implantation. The implants were fully assembled and the PET film perforated by gently touching the film with a hot solder iron tip at 550–600 °F. This resulted in ~1.5 mm diameter perforation over the primary somatosensory cortex. On the day of the experiment, mice were head-fixed in a custom treadmill under a stereo-zoom microscope under light (0.5–1%) isoflurane anesthesia. The silicone seal covering the perforation was carefully removed and the treadmill was placed under the epifluorescence microscope. A 32-channel probe (Neuronexus, A1x32-Edge-5mm-100-177-A32) was mounted on a motorized stage (MPC 385, Sutter Instruments Inc.) and guided to the center of the perforation to touch the dura at ~2.47 mm lateral and ~0.76 mm caudal from bregma. Then the See-Shell was covered with a conductive gel bath of 1% agarose and the ground electrode was placed in a corner of the gel bath. The recording probe was inserted into the brain in 10 μm steps up to a depth of 1 mm from the pial surface into the cortex using a high precision DC motor (MTS25-Z8, Thorlabs) mounted on the Sutter manipulator at a 45° entry angle.

Recordings from the neural probe were first pre-amplified (RA16PA Medusa PreAmps, Tucker Davis Technologies), then transmitted to a second amplifier and digitizer (RZ2 system, Tucker Davis Technologies). Neural data was sampled at 24 kHz and band passed at 700-5000 Hz to visualize extracellular single units or between 0 and 200 Hz to visualize local field potentials. Simultaneous mesoscale optical imaging was performed at 10 Hz. Behavior was recorded during awake trials using a high-speed camera (FL3-U3-13Y3M-C, FLIR Inc.) at 20 frames per second. At the end of the experiment, the perforation was covered with the silicone sealant.

**Data analyses**. For wide-field 1P imaging data analyses, six ROIs covering the bilateral motor, somatosensory, and visual cortices were defined. Data analysis were performed in Fiji. Average fluorescent intensity was measured for each ROI in each image. A custom MATLAB script was used to calculate the normalized change in fluorescent intensity over the time series of images. Baseline average fluorescence was obtained by averaging fluorescent intensity over the first 4 s of the time series. After normalization, the time series were filtered (2-pole Butterworth low-pass filter: 0.3 Hz)[63].

2P Ca²⁺ imaging data were analyzed with Fiji and MATLAB. Briefly, for each time series, the moco (Motion Correction) plugin[64] in Fiji was used to correct for motion artifacts. Maximum intensity and standard deviation images were used to identify cells and place ROIs over each cell in the FOV. For each ROI, the average of the pixel intensity was extracted and imported into a custom MATLAB code. Differential fluorescence intensity ($\Delta F/F_0$) was calculated for each ROI, where $F_0$ was equal to the lowest 20% of the average pixel value in the ROI over the complete time series.

Behavioral image sequences were imported and analyzed in Fiji. Each image sequence was binned at 3 × 3. ROIs were placed over the nose, forelimb, hindlimb, and the disk. Changes in average pixel intensity across the ROIs in sequential frames when there was movement detected or when there was no movement detected were given a value of 1 or 0, respectively. This allowed for various types of behavior to be quantified. Walking was classified as changes in all ROIs; grooming was classified as changes in only the forepaw and nose ROIs.

All extracellular recording data were post-processed using custom MATLAB scripts. The raw voltage traces from multiple channels were filtered using a 150th order finite impulse response filter with bounds of 800–5000 Hz. The filtered signals were thresholded to detect action potentials using previously described methods[65]. Cells were sorted using linear discriminant analyses and wavelet decomposition[65–67]. Firing rate for each cell was computed using kernel density estimation and smoothing[68]. To determine the relationship between Ca²⁺ signals and firing rates, we generated 1000 bootstrapped shuffled trials of the spike firing rate of each cell[65] and computed the cross-correlations with the Ca²⁺ activity in I-S1. The cells were categorize as modulated if their correlation coefficient at zero-lag was greater than mean + 1.96 times the standard deviation of the bootstrapped trials.

**Histology**. A subset of mice ($n = 3$ control mice and $n = 3$ mice implanted with See-Shells for 5 weeks) were fully anesthetized in 5% isoflurane and transcardially perfused with phosphate-buffered saline (PBS; CAT# P5493-1L, Sigma Aldrich) followed by 4% paraformaldehyde (PFA; CAT# P6148-500G, Sigma Aldrich). The brains were extracted and stored overnight in 4% PFA for fixation. The brain was sliced into 50 μm slices and then kept in PBS solution containing 100 mM glycine (50046-50 G, Sigma Aldrich) for 30 min to quench excess PFA. This was followed by keeping the slices in PBS solution containing 100 mM glycine and 2% Triton X-100 (93443-100 ML, Sigma Aldrich) to permeabilize the tissue. Slices were then kept in PBS solution containing Triton X-100 and blocking agent (Goat Serum, CAT# 927502, Biolegend) for 2 h after which they were incubated in the solution containing 1:1500 primary antibody–Monoclonal Anti-GFAP antibody produced in mouse (G3893-.2 ML, Sigma Aldrich) for 24 h at 4 °C. Slices were washed and incubated in solution containing the secondary antibody conjugated with fluorophores (anti-mouse Alexa 488, ab150117, abcam). Next, the slices were thoroughly washed to clear any excess antibody and mounted on glass slides in VECTASHEILD (H-1200, Vector Labs), a mounting medium containing 4′,6-diamidino-2-phenylindole (DAPI). Mounted slices were imaged under an upright confocal microscope (FV1000 BX2, Olympus FluoView). GFAP expression was quantified using a previously established protocol[22]. In each slice, average fluorescent intensity was measured using the "measure" function in Fiji at 7 ROIs (500 × 500 μm²) distributed in the image, with 4 ROIs at the pial surface and 3 ROIs in layer 2/3. This analysis was repeated in slices from three implanted and three non-surgical control mice.

## Data availability
The source data underlying Figs. 1b–d, 2c, 4d, e, 5c–e, and 6c, d and Supplementary Figs. 1, 2, 4, 6, 7, and 8 are provided as a Source Data file. All other data supporting the findings of this study are available from the corresponding author on request.

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

## Acknowledgements

S.B.K. acknowledges funds from the Mechanical Engineering department, College of Science and Engineering, MnDRIVE RSAM initiative of the University of Minnesota, McGovern Institute Neurotechnology (MINT) fund, National Institutes of Health (NIH) 1R21NS103098-01, and the Minnesota Department of Higher Education. M.L.R. was supported by 3R21 NS103098-01S1. L.G. was supported by the University of Minnesota Informatics Institute (UMII) Graduate Research fellowship. T.J.E. was supported, in part, from NIH grants RO1 NS18338 and R37 NS040389. K.W.E. acknowledges funding from DARPA grant N66001-17-2-4010. We would like to thank Bonita Van Heel at Minnesota Dental Research Center for Biomaterials and Biomechanics for help with micro-CT scanning experiments. We would also like to acknowledge Dr. Mark Sanders and Jason Mitchell at the UMN University Imaging Center where all 2P imaging experiments were conducted. We also acknowledge Dr. Jenu Chacko and Dr. Jayne Squirrell of the UW-Madison who assisted with cell culture and viability experiments for FLIM.

## Author contributions

L.G. and S.B.K. conceptualized the technology. L.G., R.E.C, T.J.E., K.W.E., and S.B.K. designed experiments. L.G., R.E.C., M.L.R., J.D., G.C., J.H., M.A.K.S., L.H., N.M., M.M.G., S.L.W., and S.B.K. conducted the experiments. L.G., R.E.C., G.C., M.A.K.S., A.N., T.J.E., and S.B.K. analyzed the data. L.G., R.E.C., M.A.K.S., T.J.E., K.W.E., and S.B.K. wrote the paper.

## Additional information

**Competing interests:** The authors declare no competing interests.

