## [Peer Review File · Nature Communications]

Reviewers' comments:

Reviewer #1 (Remarks to the Author):

Ghanbari et al. developed a transparent polymer skull replacement for exposing large areas of the dorsal mouse brain to optical neuroscience tools. Although similar in concept to Kim et al. Cell Rep 2016, this research distinguishes itself in three major ways.

1. The researchers created a 3D-printed, morphologically accurate implant system that incorporates an optically transparent PET polymer window.
2. The researchers showed implant longevity with photographs of the implant up to 36 weeks and calcium data up to 30 weeks.
3. The researchers performed simultaneous functional mesoscale calcium imaging and invasive electrical interfacing (read or write).

I recommend the manuscript to be accepted for publication after the authors address the following major comments:

1. Imaging quality of PET film compared to glass coverslip has not been adequately characterized. The researchers report 10 μm bead data, showing that the FWHM is comparable for both materials. This measure is too crude and does not translate down to diffraction-limited 2PFM. Instead, beads with sub-diffraction-limit sizes (e.g., 100 nm in diameter) should be imaged to characterize both the lateral and axial FWHM of the system point spread function. Furthermore, please make sure to report absolute intensity values obtained at the same gain setting, not individually normalized or obtained at different gain settings as in the current version.

2. Implant fabrication method unclear.

- a. Point profiling: CNC mill tip touches skull and z-coordinate registered. Does the CNC mill sense when it touches skull? Or is 'touching the skull' determined by eye?
- b. Bonding mold used to glue PET film to PMMA frame: is this 3D printed based off of skull profiling data?
- c. Does the PET film sit on top of the skull, or is it designed to insert into the craniotomy and sit on top of the brain (and below the skull edge of the craniotomy)? Please include a cross-section cartoon diagram of the entire implant on the skull.
- d. Describe method for perforating PET film for simultaneous extracellular electrophysiology (Fig. 5).

3. The CNC-guided craniotomy is an exciting engineering feat, but very little detail is reported on it. Is there a feedback mechanism to determine when skull is completely drilled through? How are differences in skull thickness across the craniotomy dealt with? A supplementary diagram and/or video would be helpful.

4. Since skull surface profiling is done only once per mouse line, the within-mouse line skull shape variance is an important metric to report, either with data or a reference.

5. "implants remained clear for several weeks after implantation, with the longest duration assessed at 36 weeks". Would like to see a statistic on the implant population. What is the surgery success rate, and what is the median lifetime of an implant? What are the reasons for termination of implant?

6. Bone regrowth is a problem common in long-term optical window implants. Optical attenuation by the bone, as well as its autofluorescence, greatly impact image quality, especially in diffraction-limited imaging. If the PET film at the edges of the craniotomy is separated from the brain surface by fluid, bone has room to regrow under the implant. It was stated "no bone regrowth was observed in any of the mice", but how was this observation made? Photographs at 30 and 36 weeks will not show bone regrowth because thin bone is nearly transparent. However, 2P imaging

quality will decline in areas of bone regrowth. Fig. s5b reports lower $\Delta F/F$ than s5a, which could be a sign of this. Furthermore, even with bone growth, there may be regions of the brain with better image quality (because the regrown bones often have large holes, through which higher quality image can be obtained). 2P imaging with a small field of view is not quite as convincing as imaging data of several brain regions across the same late-stage implant to characterize any signal degradation (or imaging data of the Thy1-YFP mouse at 30-36 weeks post implant similar to those provided in Figure 3, which were taken at 2 weeks).

7. Fig. 6 reports the across-brain activity during electric stimulation of awake and anaesthetized brain states of one mouse. Is the onset of the electric stimulation pulse train at 0 seconds?

Minor comments:

1. Fig. s2: subfigures are labeled wrong, and no dotted line drawn in A.
2. Fig. 5c: hard to see faint red raster plots.
3. Fig. 5e and Fig. 6c: please report number of averaged events.
4. In the discussion section, adapting this technology for NHPs is mentioned. What is the mechanical strength of PET compared to cranial pressures of NHPs? Is there a critical size of craniotomy that is feasible for NHPs?
5. Discussion section brings up the issue of working distance of 2P microscope objectives, but how much of an issue is this really, if small, flexible neural probes have already been used in conjunction with 2P imaging? (Ref. 72, and also Luan et al. Science Advances 2017)
6. To further the narrative that this see-shell technology can be easily adopted by other labs, price of equipment, materials, and implant fabrication time should be reported.

Reviewer #2 (Remarks to the Author):

Ghanbari and colleagues describe a method for replacing the dorsal part of the mouse cranium with a transparent PET film that allows mesoscale 2-photon laser scanning microscopy of the cerebral cortex. Using a detailed stereotactic profile of the skull surface, a 3D-printed frame was produced to which a thin PET film was bonded, which together replaced the mouse's skull. This generated a large transparent preparation through which fluorescence could be observed in transgenic mice. The authors show that the preparation is viable for many weeks. Then they demonstrate that micron-scale structures can be resolved, and single cell and neuronal population activity can be monitored.

I think this is a promising technique that could prove to be a useful addition to the existing cranial window- and large surface skull replacement techniques. However, for this paper to carry its own weight among the plethora of existing tools, the model should be characterized and described in more detail. Especially given that this is a technical and proof of principal paper, it is important to provide clearer examples and detailed information.

1. Preparation:

The configuration of the frame with the PET film is not entirely clear. At a first reading I had the impression that the stereotactic coordinates were taken to print a 3D mold on which the PET film was coated, but later realized that the PET film is force-bond to the frame by a mold. It looks like that this mold as depicted in Supl. Fig. 3 has the shape of a skull, but it is based on the 3D coordinates? When the film is force-bond from a flat sheet one expects stretching artifacts and perhaps non-homogeneous optical properties across the skull.

Related to this, it is not clear what was the typical thickness of the PET film, and how this was adjusted to match the skulls inner surface. It is mentioned that the frame was extruded by 0.6 to

0.8 mm. Was this done for reasons to compensate for the loss of the thickness once the skull is removed? I think this could be explained in more detail. In general, one would like to have the film take the shape of the brain rather than the outside profile of the skull. Could one use a mold based on a brain rather than skull?

If the coordinates were only used for 3D printing of the frame, I do not exactly get the rationale of mapping more than 80 coordinates over the entire surface.

It should be made clear at the beginning of the paper that the frame is used for multiple mice. At first reading (or based on the summary) it may look as if a print is made for each mouse.

2. Optical characterization:

It is not clear how the authors conclude that the spatial resolution through this prep is similar to glass. The transmission data are useful but does not allow drawing conclusions as to the resolution through this material.

Fig 3 looks quite impressive, but it doesn't give the actual resolution. Fig. 1b is blurry, and based on large beads. The authors should compare glass and PET with a high magnification objective and very small beads and provide a point spread function for images through both materials.

The rationale of the FLIM experiment is unclear. Are they measuring FLIM of PET or of fluorescent material through PET? Why? This experiment is not supported by any figure.

3. Characterization of viability

The panels in figure 2C are mixed up. DAPI seems to denote the GFAP staining and vice versa. The cortical areas in the experimental and control mice seem different. In addition, we don't see clear GFAP-positive cells (except maybe in the hippocampus). What is quantified exactly? The authors should give clearer examples and quantify this more thoroughly.

It is surprising that there is no bone regrowth. The authors must have experienced complications of some sorts. Can the authors provide more details on what type of growth they observe, or what are the potential pitfalls with this technique? What is success percentage etc?

4. Details of the ephys are missing. How was the clustering done? Electrodes etc? The main text and methods seem not congruent with one another at this point.

5. Longitudinal imaging data are lacking. While the authors show that See-Shells allows for cellular resolution imaging in the same mouse for multiple experimental sessions across several months (Supplementary Figure 5b), they don't give examples of single neurons monitored longitudinally. This would showcase the true potential of the technique and strengthen the paper.

We would like to thank the reviewers for constructive feedback on the manuscript. We have significantly revised the manuscript, and hope these address the reviewer and editorial concerns adequately. In addition to reviewer comments, we have also made minor changes to the manuscript to comply with editorial policies. All edited and new text in the manuscript are in blue.

Major changes to the manuscript include new data showing the comprehensive characterization of the optical resolution of PET and comparison with a glass coverslip, data demonstrating longitudinal imaging of neuronal structures as well as calcium activity of single neurons, assessment of calcium imaging quality during late-stage implantation, and documentation on the longevity of the implants. We hope these additions fully address the concerns of the reviewers.

Reviewers' comments:

Reviewer #1 (Remarks to the Author):

Ghanbari et al. developed a transparent polymer skull replacement for exposing large areas of the dorsal mouse brain to optical neuroscience tools. Although similar in concept to Kim et al. Cell Rep 2016, this research distinguishes itself in three major ways.

- 1. The researchers created a 3D-printed, morphologically accurate implant system that incorporates an optically transparent PET polymer window.*
- 2. The researchers showed implant longevity with photographs of the implant up to 36 weeks and calcium data up to 30 weeks.*
- 3. The researchers performed simultaneous functional mesoscale calcium imaging and invasive electrical interfacing (read or write).*

I recommend the manuscript to be accepted for publication after the authors address the following major comments:

We thank the reviewer for a succinct description of the technological advancements and the recommendation for publication.

1. Imaging quality of PET film compared to a glass coverslip has not been adequately characterized. The researchers report 10 μm bead data, showing that the FWHM is comparable for both materials. This measure is too crude and does not translate down to diffraction-limited 2PFM. Instead, beads with sub-diffraction-limit sizes (e.g., 100 nm in diameter) should be imaged to characterize both the lateral and axial FWHM of the system point spread function.

This is a very good point made by the reviewer. We now include new data in the revised **Figure 1** showing the point spread functions (PSFs) of sub-diffraction limited 200 nm YG beads imaged with a high magnification imaging objective. See **Page 3, lines 127 - 136** for new results:

*“Sub-diffraction limit 200 nm fluorescent beads were imaged using a high magnification (40x) objective through glass coverslip and PET film to construct the lateral and axial point spread functions (PSFs, **Fig. 1b** and **c**). For beads imaged through PET the full width at half maximum (FWHM) of the lateral PSF (**Fig. 1b**) was 425 ± 26.2 nm and was 409.3 ± 13.7 nm for beads imaged through glass coverslip. No significant difference was observed between PET and coverslip glass (**Fig. 1d top**, $p = 0.35$, Welch's t-test). For beads imaged through PET the FWHM for the axial PSF (**Fig. 1c**) was 2.88 ± 0.08 μm and was $3.01 \pm$*

0.10 μm for beads imaged through glass coverslip. Again, no significant difference was observed (**Fig. 1d bottom**, $p = 0.34$, Welch's *t*-test)."

Furthermore, please make sure to report absolute intensity values obtained at the same gain setting, not individually normalized or obtained at different gain settings as in the current version.

We have now reported absolute intensity values.

2. Implant fabrication method unclear.

We have now made significant changes to the manuscript to better articulate this aspect of the manuscript. In particular, please see modified **Supplementary Figure 3** that includes an updated CAD schematic and new photographs of the implant assembly process.

a. Point profiling: CNC mill tip touches skull and z-coordinate registered. Does the CNC mill sense when it touches skull? Or is 'touching the skull' determined by eye?

The probe contact with the skull was determined by eye. We have modified the methods section describing this process at **Page 11 lines 449 - 453**:

"The end mill mounted on a motorized stage was carefully lowered until the end mill tip made contact with the skull surface at bregma. This process was visualized at the highest magnification setting of the stereo-zoom microscope (6x, M60, Leica) to ensure the tip did not exert a force sufficient to deform the skull surface before registering the coordinates in the LabVIEW program."

We note that probe contact with the skull has now been fully automated, via a low force contact sensor and is described in a manuscript just been accepted for publication (Ghanbari* Rynes* et al., Scientific Reports). A preprint of the paper can be accessed at: <https://www.biorxiv.org/content/early/2018/11/29/480004>).

b. Bonding mold used to glue PET film to PMMA frame: is this 3D printed based off of skull profiling data?

Yes. The bonding mold is 3D printed based off of the skull profiling data.

c. Does the PET film sit on top of the skull, or is it designed to insert into the craniotomy and sit on top of the brain (and below the skull edge of the craniotomy)? Please include a cross-section cartoon diagram of the entire implant on the skull.

The PET film sits on top of the skull. We have included a cross-sectional cartoon diagram illustrating this. See **Supplementary Fig. 3g**.

d. Describe method for perforating PET film for simultaneous extracellular electrophysiology (Fig. 5).

We have included these details in the Methods section. See **Page 15 lines 635 - 638**:

"The implants were fully assembled and the PET film perforated by gently touching the film with a hot solder iron tip at 550 – 600° F. This resulted in ~1.5 mm diameter perforation over the primary somatosensory cortex."

3. The CNC-guided craniotomy is an exciting engineering feat, but very little detail is reported on it. Is there a feedback mechanism to determine when skull is completely drilled through? How are differences in skull thickness across the craniotomy dealt with? A supplementary diagram and/or video would be helpful.

We thank the reviewer for the comment. We have described the craniotomy robot in great detail in a separate manuscript accepted for publication (Ghanbari* Rynes* et al., Scientific Reports). A preprint of the paper can be accessed at:

<https://www.biorxiv.org/content/early/2018/11/29/480004>).

For the sake of completeness, the revised manuscript includes a brief description of this process while citing the original work. See **Page 13 lines 538 - 551**:

“The CNC milling machine incorporated in the stereotaxic instrument was used to perform automated craniotomies in C57BL/6, Thy1-YFP (#003709, Jackson Laboratories), Thy1-GFPm (#007788, Jackson Laboratories) and Thy1-GCaMP6f mice (#024276, Jackson Laboratories) (Supplementary Figure 1a) as described previously[40]. Briefly, the CNC end mill was lowered to the skull surface at pilot points along a predefined path slightly smaller than the perimeter of the See-Shell. Once the contact was confirmed visually through a stereomicroscope, the z-coordinate at that point was registered. This process was repeated at multiple points along the desired craniotomy path. The registered coordinates were then used to interpolate a 3D cutting path for milling the skull in the LabVIEW program. For each craniotomy, the initial milling depth was 50 μm , which was well within the thickness of the skull. In each subsequent milling pass, the depth was incremented in 10 μm steps until reaching the soft part of the bone or trabeculae in a section of the craniotomy path. This was sufficient to pry open the bone for excision across the whole craniotomy.”

4. Since skull surface profiling is done only once per mouse line, the within-mouse line skull shape variance is an important metric to report, either with data or a reference.

This is a very good point made by the reviewer. We have now included two references that support our reasoning that the surface profile from a single mouse at a single time point during adulthood is sufficient to design a generalized polymer skull for that strain.

First, a cranial morphometry study of commonly used inbred laboratory mouse strains (Kawakami and Yamamura, BMC Evolutionary Biology 2008) found that intra-species variations in cranial bone shape and size are minimal. For instance, in this study, the length and width of the frontal bone of C57BL/6 mice were reported to be 7.756 +/- 0.073 mm and 5.356 +/- 0.101 mm respectively. The length and width of the parietal bone were reported to be 3.889 +/- 0.162 mm and 8.1 +/- 0.229 mm respectively. Thus, variances in skull bones sizes within a strain are in the order of only tens to a couple of hundred micrometers.

We also note findings from the second study where post-natal size and shape of the skull were tracked over time (Vora et al., Frontiers in Physiology 2016). This study showed that both the width and length of the cranial bones above the dorsal skull of C57BL/6 mice are established in the first 3 weeks and there is no significant variation during adulthood. Our implants were designed using surface profile obtained from a mouse that was 8 weeks old (C57BL/6) and 16 weeks old (*tg/tg*). The implants derived from these profiles were all implanted on mice of similar age or older.

As a final note, Thy1-GCaMP6f, Thy1-YFP and Thy1-GFPm mice are derived from C57BL/6 lines. Thus, implants based on the surface profile from the C57BL/6 mouse readily fit these transgenic mice. Edits in response to this comment are now in **Page 3 lines 112 - 118**:

“Previous cranial morphometry studies of commonly used inbred laboratory mouse strains have shown that intra-species variations in cranial bone shape and size are minimal[16]. Further, postnatal size and shape of the skull are established within the first 3 weeks and change minimally after reaching adulthood[17]. Thus, the interpolated surface from a single mouse skull served as a template to digitally design generalized transparent skulls (“See-Shells”) using computer-aided design (CAD) software.”

*5. “implants remained clear for several weeks after implantation, with the longest duration assessed at 36 weeks”. Would like to see a statistic on the implant population. What is the **surgery success rate**, and what is the **median lifetime of an implant**? What are the reasons for termination of implant?*

We have now included the following details in surgical success rate, the median lifetime of the implant and reasons for termination of an implant.

Surgery success rate: Surgeries were performed by 3 separate surgeons, all co-authors of the manuscript. Some surgeries were terminated via euthanasia without implantation of the device if the dura was accidentally torn or damaged during skull excision or if brain edema was observed. Surgeries were also terminated via euthanasia if we observed deviations from normal physiology under anesthesia (e.g., irregular breathing) as required by our IACUC protocol. These experiments have not been included in this study.

Median lifetime of implant: We have now reported the median lifetime of the implants.

Reasons for termination of implants: 3 mice with chronic implants were found to have improper implantations (e.g., loose dental cement) and the mice were euthanized within 10 days. A subset of the mice (n = 3 mice) was perfused and brains dissected after 4 weeks for histological analysis. Subsets of the animals were used to calcium imaging, structural imaging, simultaneous mesoscale imaging and electrophysiology, and mesoscale imaging during cortical microstimulation. These were typically euthanized after necessary experiments were conducted (2-16 weeks after implantation). A subset of the mice (n = 24) was used to assess the longevity of the implants. The results of these evaluations are now described in the modified manuscript. Please see **Page 4 lines 163 - 170**:

*“See-Shells were chronically implanted on wild-type C57BL/6 (n = 9), Thy1-GCaMP6f (n = 31), Thy1-YFP mice (n = 3) and Thy1-GFPm mice (n = 3) after a craniotomy was performed to remove the skull over the dorsal cortex (**Fig. 2a**, see **Methods**). The median duration of the implantation was 92 days, with durations ranging from 7 days to 337 days. Of these, some procedures failed because the dental cement used to seal the implants (see **Methods**) had not sufficiently adhered to the skull. These experiments were terminated within 10 days of surgery. Thus, the overall surgery success rate was 93.5% (3 surgeons, n = 43/46 mice).”*

6. Bone regrowth is a problem common in long-term optical window implants. Optical attenuation by the bone, as well as its autofluorescence, greatly impact image quality, especially in diffraction-limited imaging. If the PET film at the edges of the craniotomy is separated from the brain surface by fluid, bone has room to regrow under the implant. It was stated “no bone regrowth was observed in any of the mice”, but how was this observation made?

We originally assessed bone regrowth via visual observation under a stereozoom microscope.

Photographs at 30 and 36 weeks will not show bone regrowth because thin bone is nearly transparent. However, 2P imaging quality will decline in areas of bone regrowth. Fig. s5b reports lower $\Delta F/F$ than s5a, which could be a sign of this. Furthermore, even with bone growth,

there may be regions of the brain with better image quality (because the regrown bones often have large holes, through which higher quality image can be obtained). 2P imaging with a small field of view is not quite as convincing as imaging data of several brain regions across the same late-stage implant to characterize any signal degradation (or imaging data of the Thy1-YFP mouse at 30-36 weeks post implant similar to those provided in Figure 3, which were taken at 2 weeks).

The reviewer has brought up an important point. We have performed new experiments and present new data.

Specifically,

1) We have now imaged several locations randomly accessed across the whole cortex in a Thy1-GCaMP6f mouse 335 days after implantation. As shown in **Supplementary Figure 7**, maximum intensity projections of the acquired time series show that single neurons and their activities can be clearly discriminated in all the areas imaged. Analyzing the activities of randomly sampled neurons in these fields of view, we found that the peak $\Delta F/F$ values during spontaneous head fixed behavior ranged between 61.51% to 191.84%.

2) We have also presented new data showing the *same* neurons imaged in multiple sessions over a 4-week period several months after implantation. (307 – 335 days after implantation) (**Figure 4**). Peak $\Delta F/F$ of each individual neuron assessed was >98% in all the imaging sessions. Linear regression of peak $\Delta F/F$ values neurons sampled across imaging sessions was indicated a very little decrease in signal quality.

These new data are presented in **Page 6 lines 243 - 255**:

*“To evaluate See-Shells’ capability to monitor Ca^{2+} signals in the same neurons over time, multiple imaging sessions were performed in the same FOV over a month starting 44 weeks after implantation on a Thy1-GCaMP6f mouse (**Fig. 4e**). Average intensity projections from a set of high-resolution images qualitatively indicated that the same neurons could be identified over time. Robust Ca^{2+} signals were obtained with the peak $\Delta F/F$ of randomly selected individual neurons ranging between 128.7% to 240.4% across all imaging sessions. Linear regression analysis of peak $\Delta F/F$ measured over imaging sessions indicated that Ca^{2+} signals were diminished slightly across duration evaluated ($R^2 = 0.378$, average slope of trend-line = -1.23% per day, $n = 5$ neurons, **Supplementary Figure 6**). Further, 2P imaging was performed at multiple sites distributed across the cortex in a late stage implanted mouse (Day 335, **Supplementary Figure 7**). Similar to the data shown in **Figure 4**, robust Ca^{2+} signals were acquired from each region with peak $\Delta F/F$ of individual neurons ranging between 61.51% to 191.84%.”*

Please also see revised **Figure 4, Supplementary Figures 6 and 7**.

3) Finally, we re-examined 24 mice for which we have wide-field images taken at multiple time points after implantation. We did observe opacity of the implants in 4 of the mice some part of the imaging field. We have now documented these findings in the new manuscript. We have also toned down this section in the manuscript and report the median duration of implants.

These results are described in **Page 4 lines 170 - 176**:

“A subset of the mice was observed under a high magnification (6x) stereo-zoom microscope to assess implant opacity or bone regrowth. In 75% of the mice ($n = 18/24$) no opacity of the windows or bone regrowth was observed in any part of the field of view (FOV). In 3 mice, significant opacity (10% - 30% of the FOV) was observed within 60 days. In 3 mice, opacity blocking optical access to <10% of the FOV, along the midline suture was observed after >100 days of implantation with 48 weeks being the longest duration assessed.”

7. Fig. 6 reports the across-brain activity during electric stimulation of awake and anaesthetized brain states of one mouse. Is the onset of the electric stimulation pulse train at 0 seconds?

The dotted line in **Fig 6c** indicates the onset of the stimulation. Further, times indicated in **Fig. 6b** are post-stimulation, as clarified in the modified figure legend.

Minor comments:

1. Fig. s2: subfigures are labeled wrong, and no dotted line drawn in A.

We thank the reviewer for pointing this out. The labels have been corrected in the revised manuscript.

2. Fig. 5c: hard to see faint red raster plots.

We thank the reviewer for pointing this out and the problem has been fixed in the revised manuscript.

3. Fig. 5e and Fig. 6c: please report number of averaged events.

Fig 5e: The averaged number of events is 1000. it is now reported (**Page 6 line 290**).

Fig 6c: We have now reported the number of averaged events (**Page 7 lines 305 - 308**).

4. In the discussion section, adapting this technology for NHPs is mentioned. What is the mechanical strength of PET compared to cranial pressures of NHPs? Is there a critical size of craniotomy that is feasible for NHPs?

Previous studies report that the maximum intracranial pressure in non-human primates (NHP) is 9.8 KPa (Gucer et al J Neurosurgery 1979). The Young's modulus and ultimate tensile strength of PET (as reported in engineering datasheets) are 2-2.7 MPa and 55 MPa, which are several orders of magnitude higher than NHP intracranial pressure. Thus, the intracranial pressure is unlikely to cause significant strain/stretching of the PET. Further, it is hard to comment on the critical size of the implant as it will be a function of several factors including the physiology of the NHP brain and any estimates we come up with, without performing a thorough study will be purely speculative. This is indeed the reason why we mention this as a potential future application in the discussion section of the manuscript.

*5. Discussion section brings up the issue of **working distance of 2P microscope objectives**, but how much of an issue is this really, if small, flexible neural probes have already been used in conjunction with 2P imaging? (Ref. 72, and also Luan et al. Science Advances 2017)*

It is indeed true that flexible neural probes have already been used in conjunction with 2P imaging. We thank the reviewer for pointing out the original reference, which is now included as a reference. However, the vast majority of neuroscientists still utilize commercially available silicon-based neural probes. Until ultra-flexible probes gain traction in the community, it is advantageous to develop objectives that are compatible with such rigid neural probes, so that our technology to have the broadest applicability. Such objectives could be useful for combining other modalities too, such as fibers for optogenetic stimulation.

6. To further the narrative that this see-shell technology can be easily adopted by other labs, price of equipment, materials, and implant fabrication time should be reported.

This is an important point brought up by the reviewer. We have reported the price in the main manuscript. According to our estimates, it costs less than \$20 to fabricate and takes < 15 minutes to assemble (**Page 8 lines 366 - 367**). We have also modified the **Methods** section to include part numbers.

Reviewer #2 (Remarks to the Author):

Ghanbari and colleagues describe a method for replacing the dorsal part of the mouse cranium with a transparent PET film that allows mesoscale 2-photon laser scanning microscopy of the cerebral cortex. Using a detailed stereotactic profile of the skull surface, a 3D-printed frame was produced to which a thin PET film was bonded, which together replaced the mouse's skull. This generated a large transparent preparation through which fluorescence could be observed in transgenic mice. The authors show that the preparation is viable for many weeks. Then they demonstrate that micron-scale structures can be resolved, and single cell and neuronal population activity can be monitored.

I think this is a promising technique that could prove to be a useful addition to the existing cranial window- and large surface skull replacement techniques. However, for this paper to carry its own weight among the plethora of existing tools, the model should be characterized and described in more detail. Especially given that this is a technical and proof of principal paper, it is important to provide clearer examples and detailed information.

We thank the reviewer for the encouraging comments. We have now performed several new experiments that provide clearer examples and more detailed information.

1. Preparation:

The configuration of the frame with the PET film is not entirely clear. At a first reading I had the impression that the stereotactic coordinates were taken to print a 3D mold on which the PET film was coated, but later realized that the PET film is force-bond to the frame by a mold. It looks like that this mold as depicted in Supl. Fig. 3 has the shape of a skull, but it is based on the 3D coordinates?.

Yes, the mold is based on the shape of the skull.

When the film is force-bond from a flat sheet one expects stretching artifacts and perhaps non-homogeneous optical properties across the skull

The PET has relief cuts to prevent it from plastic deformation during the bonding process and the clamping is done gently and for a short duration of time (5-10 minutes) to ensure adequate contact between the PET film and the PMMA frame. The PET film is sandwiched between the PMMA frame and molds during this process. While it bends and conforms to the shape of the mold, there is no plastic deformation or stretching. Thus, the bonding procedure does not result in non-homogenous optical properties. **Supplementary Figure 3** has been modified to better illustrate these aspects.

Related to this, it is not clear what was the typical thickness of the PET film, and how this was adjusted to match the skulls inner surface. It is mentioned that the frame was extruded by 0.6 to 0.8 mm. Was this done for reasons to compensate for the loss of the thickness once the skull is removed? I think this could be explained in more detail.

The PET film is 50 μm thick. 'Extrusion' refers to a function in the computer-aided design software that allows the surface interpolated from the pilot points to have the thickness of the 3D printed PMMA frame illustrated in **Figure 1**. The thickness of 600-800 μm was used to ensure sufficient structural integrity to the frame, so other components such as the PET film and the headplate, could be integrated into the implant. It was not done to compensate for the loss of

the thickness of the skull. We have modified **Supplementary Figure 3** to incorporate a cartoon schematic of the implanted device. We hope this illustrates the implant-skull-brain interface better.

In general, one would like to have the film take the shape of the brain rather than the outside profile of the skull. Could one use a mold based on a brain rather than skull?

In response, we make three points:

1) As shown in coronal sections of the micro-CT scan of a C57BL/6 mouse below, the thickness of the skull over the dorsal cortex is uniform for most of the cross-section. Apart from the area within ~0.5mm from the midline suture, the lower surface of the skull runs parallel to the top surface, offset by 50-200 μm for much of the cross-section. Thus, the skull's top surface is a good approximation of the bottom surface of the skull or the brain.

Coronal section 2 mm posterior to Bregma

Coronal section at Bregma

2) A mold based on the brain will be difficult to generate because of the problems with profiling the brain surface that is considerably softer than the skull and more prone to being damaged.

3) Information of the skull's top surface allows us to design an implant that sits on top of the remaining skull at the edge of the craniotomy. Thus, a hermetic seal can be achieved during the implantation using cyanoacrylate glue and dental cement application. See **Supplementary Figure 3g** for a cartoon illustration of the implant-skull-brain interface.

If the coordinates were only used for 3D printing of the frame, I do not exactly get the rationale of mapping more than 80 coordinates over the entire surface.

Coordinates (3D point cloud) were used to interpolate the 3D surface of the skull. This surface was used as a template to design the See-Shell frame, as well as two shaping molds that curve the PET film during the bonding process (modified **Supplementary Fig. 3**). It is possible we could have generated these components with a lesser number of points. We did not systematically explore/optimize this step, as a profile from a single mouse was sufficient to generate implants for the two mouse lines studied.

It should be made clear at the beginning of the paper that the frame is used for multiple mice. At first reading (or based on the summary) it may look as if a print is made for each mouse.

We have modified the results section to clearly articulate this point. See **Page 3 lines 112 - 118**:

“Thus, the interpolated surface from a single mouse skull served as a template to digitally design generalized transparent skulls (“See-Shells”) using computer-aided design (CAD) software.”

2. Optical characterization:

It is not clear how the authors conclude that the spatial resolution through this prep is similar to glass. The transmission data are useful but does not allow drawing conclusions as to the resolution through this material. Fig 3 looks quite impressive, but it doesn't give the actual resolution. Fig. 1b is blurry, and based on large beads. The authors should compare glass and PET with a high magnification objective and very small beads and provide a point spread function for images through both materials.

This is a very good point made by the reviewer. We now include new data in the revised **Figure 1** showing the point spread functions (PSFs) of sub-diffraction limited 200 nm YG beads imaged with a high magnification imaging objective. See **Page 3 lines 127 - 136** for new results:

*“Sub-diffraction limit 200 nm fluorescent beads were imaged using a high magnification (40x) objective through glass coverslip and PET film to construct the lateral and axial point spread functions (PSFs, **Fig. 1b** and **c**). For beads imaged through PET the full width at half maximum (FWHM) of the lateral PSF (**Fig. 1b**) was 425 ± 26.2 nm and was 409.3 ± 13.7 nm for beads imaged through glass coverslip. No significant difference was observed between PET and coverslip glass (**Fig. 1d top**, $p = 0.35$, Welch's t-test). For beads imaged through PET the FWHM for the axial PSF (**Fig. 1c**) was 2.88 ± 0.08 μ m and was 3.01 ± 0.10 μ m for beads imaged through glass coverslip. Again, no significant difference was observed (**Fig. 1d bottom**, $p = 0.34$, Welch's t-test).”*

The rationale of the FLIM experiment is unclear.

Fluorescent lifetime imaging (FLIM) is commonly used to visualize and quantify endogenous fluorophores such as NADH/FAD in studies of metabolic activity. FLIM imaging of NADH/FAD is particularly useful for studying tumor micro-environments. Given the flexibility of See-Shells and the ability to generate morphologically accurate implants, there is the potential for intra-vital FLIM imaging of different organs.

Are they measuring FLIM of PET or of fluorescent material through PET?

We are measuring fluorescence lifetimes of the beads when imaged through the PET film. As FLIM is very sensitive to environment and specimen preparation, we performed these experiments to test if imaging through the PET causes any changes to FLIM measurements.

This experiment is not supported by any figure.

We have now summarized this data in **Supplementary Figure 4**.

3. Characterization of viability. The panels in figure 2C are mixed up. DAPI seems to denote the GFAP staining and vice versa.

We thank the reviewer for pointing this out and we have now fixed this error. Apologies for the oversight.

The cortical areas in the experimental and control mice seem different.

All images were taken from the same cortical areas: ~ 2.0 mm posterior to Bregma and 1.5 mm lateral to the midline. The images have been taken from slices mounted at slightly varying

angles which may have led to this confusion. We report this information in the revised methods section.

In addition, we don't we see clear GFAP-positive cells (except maybe in the hippocampus).

Indeed, we do not see GFAP labeled cells in the cortex. This is consistent with previous studies analyzing GFAP expression in the mouse brain. See GENSAT:

<http://www.gensat.org/imagenavigator.jsp?imageID=14204> (Gong et al. Nature 2003).

In the uninjured brain, the expectation is to observe cortical astrocytes expressing GFAP in the corpus callosum, the brain surface and at the regions bordering the ventricles. Our observations were as expected in both control and implanted mice.

What is quantified exactly? The authors should give clearer examples and quantify this more thoroughly.

We assessed the overall concentration of the GFAP protein in the images. In mice, GFAP concentration is correlated with the total number of astrocytes and in regions where low astrocyte numbers are seen (e.g., cortex and striatum), measuring the fluorescent intensity of GFAP allows quantitative assessment of neuro-inflammation (Martin et al., J Neuroscience Methods 1995). As we did not observe activated astrocytes in the cortex, we utilized previously established protocols (Cvetanovic et al., Neuroscience 2015) to quantify the level of GFAP staining. The revision includes the details in the methods section and the citations. We also note that our results are consistent with those reported for implanted curve glass windows (Kim et al., 2016 Cell Reports 2016). **See Page 5 lines 190 - 196:**

"No activated astrocytes were observed in the cortex of the implanted and control mice consistent with previous studies[21]. To assess if there is any increased expression of GFAP, the fluorescent intensity was measured in multiple cortical areas using the methodology described previously[22]. GFAP fluorescence intensity in arbitrary units (a.u.), in the areas assessed, was 15.26 ± 1.64 a.u. in the implanted mice. In comparison, GFAP fluorescence intensity was 14.55 ± 1.99 a.u. in naïve control mice."

It is surprising that there is no bone regrowth. The authors must have experienced complications of some sorts. Can the authors provide more details on what type of growth they observe, or what are the potential pitfalls with this technique? What is success percentage etc?

We point the reviewer to the detailed response to points 5 and 6 raised by **Reviewer #1**. Quantification and detailed discussion of these issues are reported in the revised manuscript.

4. Details of the ephys are missing. How was the clustering done?

Our electrophysiology data analyses were performed using custom code written in MATLAB and were based on previously published methods. A thresholding algorithm (Quirogi et al., Neural Computations 2004) along with 4th order Harr wavelet transform was used to obtain wavelet coefficients. Linear Discriminant Analysis with Gaussian Mixture Model corrections (Yang et al., NIPS 2018) were used to obtain coefficients that represented the action potential waveforms. Other coefficients obtained included PCA coefficients as well as first/second derivatives (for which there is a lot of evidence as effective metrics to use). Once the coefficients were obtained, a modified KS-test (see Quiroga et al., Neural Computations 2004) was used to test the distributions for normality. Clustering, based on comparisons of two various different coefficients, was then performed manually.

Electrodes etc? The main text and methods seem not congruent with one another at this point.

A 32-channel Neuronexus probe (A1x32-Edge-5mm-100-177-A32) was used for the electrophysiology recordings. We have now made changes to the manuscript to articulate this better.

5. Longitudinal imaging data are lacking. While the authors show that See-Shells allows for cellular resolution imaging in the same mouse for multiple experimental sessions across several months (Supplementary Figure 5b), they don't give examples of single neurons monitored longitudinally. This would showcase the true potential of the technique and strengthen the paper.

We thank the reviewer for this suggestion. We have now performed extensive new experiments showcasing the longitudinal imaging capabilities. Specifically, we include Ca^{2+} imaging data from the same neurons over multiple imaging sessions spread over 1 month in a Thy1-GCaMP6f mouse (see modified **Figure 4e** and **Supplementary Figures 6 and 7**). The revised **Figure 3**, longitudinal structural imaging of the same dendrites and dendritic spines in Thy1-GFPm mice have been included.

REVIEWERS' COMMENTS:

Reviewer #1 (Remarks to the Author):

The authors have addressed all my concerns. Well done!

Reviewer #2 (Remarks to the Author):

I think the revised manuscript looks much better. I don't see any obstacles for publication.

REVIEWERS' COMMENTS:

Reviewer #1 (Remarks to the Author):

The authors have addressed all my concerns. Well done!

Reviewer #2 (Remarks to the Author):

I think the revised manuscript looks much better. I don't see any obstacles for publication.

Based on the comment above, we have not made any changes in response to the reviewer comments.